# Tenotomy-induced muscle atrophy is sex-specific and independent of NFκB

Gretchen A Meyer[1,2,3]*, Stavros Thomopoulos[4], Yousef Abu-Amer[2,5,6], Karen C Shen[1]

[1]Program in Physical Therapy, Washington University School of Medicine, St. Louis, United States; [2]Department of Orthopaedic Surgery, Washington University School of Medicine, St Louis, United States; [3]Departments of Neurology and Biomedical Engineering, Washington University School of Medicine, St. Louis, United States; [4]Departments of Orthopaedic Surgery and Biomedical Engineering, Columbia University, New York, United States; [5]Department of Cell Biology & Physiology, Washington University School of Medicine, St. Louis, United States; [6]Shriners Hospital for Children, St. Louis, United States

**\*For correspondence:**
meyerg@wustl.edu

**Abstract** The nuclear factor-κB (NFκB) pathway is a major thoroughfare for skeletal muscle atrophy and is driven by diverse stimuli. Targeted inhibition of NFκB through its canonical mediator IKKβ effectively mitigates loss of muscle mass across many conditions, from denervation to unloading to cancer. In this study, we used gain- and loss-of-function mouse models to examine the role of NFκB in muscle atrophy following rotator cuff tenotomy – a model of chronic rotator cuff tear. IKKβ was knocked down or constitutively activated in muscle-specific inducible transgenic mice to elicit a twofold gain or loss of NFκB signaling. Surprisingly, neither knockdown of IKKβ nor over-expression of caIKKβ significantly altered the loss of muscle mass following tenotomy. This finding was consistent across measures of morphological adaptation (fiber cross-sectional area, fiber length, fiber number), tissue pathology (fibrosis and fatty infiltration), and intracellular signaling (ubiquitin-proteasome, autophagy). Intriguingly, late-stage tenotomy-induced atrophy was exacerbated in male mice compared with female mice. This sex specificity was driven by ongoing decreases in fiber cross-sectional area, which paralleled the accumulation of large autophagic vesicles in male, but not female muscle. These findings suggest that tenotomy-induced atrophy is not dependent on NFκB and instead may be regulated by autophagy in a sex-specific manner.

## Editor's evaluation

This articlce challenged the premise that NF-kappaB and its upstream kinase IKKbeta play a role in muscle atrophy following tenotomy. Two animal models were used – one leading to enhanced muscle-specific NF-kappaB activation and the other a muscle-specific deletion. In both models, there was no significant relationship to observed muscle changes following tenotomy. Overall, this work is significant in that it challenges the existing dogma that NF-kappaB plays a crucial role in muscle atrophy.

## Introduction

At the most basic level, skeletal muscle atrophy is a shift in the physiological balance between protein synthesis and breakdown. This shift can be initiated by a wide variety of stimuli, from disuse to starvation to cancer, but all eventually converge on a common result – reduced muscle mass. As expected from the diverse stimuli, atrophy can be initiated through a variety of intracellular pathways. However,

**eLife digest** Muscle atrophy – the gradual loss of muscle mass – follows injuries to our muscles, tendons, or joints. During atrophy, muscles shrink and become weaker, which can interfere with everyday activities and, ultimately, decrease quality of life.

Rotator cuff tears are a common example of such injuries. A rotator cuff is group of four muscles that come together as tendons to form a cuff that normally stabilises our shoulders and allows us to lift and move our arms over our heads. Rotator cuff tears can result from an injury or may be caused by ageing-related wear and tear of the tendon.

A signalling protein, called NFκB, is thought to be involved in muscle atrophy. When the NFκB signal is switched on, it interacts with genes that are thought to speed up the loss of muscle mass. However, NFκB's precise role in atrophy and recovery after muscle injury is still poorly understood, particularly following injuries where a tendon is cut or torn. Meyer et al. therefore set out to determine whether or not NFκB played a role in the muscle atrophy following rotator cuff tears.

Meyer et al. used genetically engineered mice in which NFκB's signal could be turned off at the time of rotator cuff injury, and specifically in muscle cells (but not other parts of the body). The experiments revealed that stopping NFκβ signalling in these mice did not reduce muscle atrophy after a rotator cuff injury: the levels of atrophy, muscle performance, and muscle composition were the same regardless of whether the NFκβ signal was active.

The sex of the mice did, however, affect muscle atrophy, specifically the way in which they lost muscle mass. In male mice, the size of muscle cells decreased, while in female mice, the number of muscle cells decreased. Muscle cells in male mice (but not in females) also accumulated abnormally high amounts of protein, which is an indication of a mechanism of muscle breakdown called autophagy.

These results shed new light on the way that we lose muscle mass after injury, and how that could vary depending on the individual. Meyer et al. hope that this study will help guide the development of new, more effective treatments for muscle atrophy, and ultimately contribute to therapies tailored to the characteristics of the patient and the type of injury.

studies have found surprising commonality downstream, leading to the identification of a number of 'lynchpin' mediators – in particular signaling through nuclear factor kappa beta (NFκB), which has been described as both necessary and sufficient for skeletal muscle atrophy (*Jackman et al., 2013*; *Sandri, 2013*).

Our current understanding of NFκB's role in adult muscle atrophy has been reviewed in detail (*Jackman et al., 2013*), where the reader can find illustrations of the pathway. In brief, NFκB is a family of transcription factor subunits that exist as dimers and are localized to the cytoplasm by inhibitory kinases (inhibitor of κB, IκB). Upon stimulation, IκB kinases (IKKα and IKKβ) are phosphorylated and in turn phosphorylate IκB (predominantly IκBα). This leads to cytosolic degradation of Iκβ and nuclear transport of NFκB transcription factor dimers to initiate transcription of genes involved in the atrophy program. Inhibiting IKKβ activity by muscle-specific knockout of its gene (*Ikbkb*) or electroporating a dominant negative form into skeletal muscle prevented approximately half the loss of muscle mass induced by denervation (*Mourkioti et al., 2006*), unloading (*Van Gammeren et al., 2009*), or immobilization (*Reed et al., 2011*). Similar results were found in the Iκβα superrepressor mouse, which displays dampened NFκB activity, in response to an inflammatory insult (*Haegens et al., 2012*; *Langen et al., 2012*) denervation (*Cai et al., 2004*), unloading (*Judge et al., 2007*), nutrient deprivation (*Lee and Goldberg, 2015*), or tumor (*Cai et al., 2004*). Assessment of NFκB activity by nuclear localization or DNA binding of its subunits confirmed that inhibition of IKKβ activity or IκBα degradation completely (*Cai et al., 2004*; *Senf et al., 2008*; *Van Gammeren et al., 2009*) or partially (*Mourkioti et al., 2006*) blocked NFκB translocation, demonstrating that NFκB signaling is responsible for a considerable portion of atrophy driven by these diverse stimuli.

The best-studied gene targets of NFκB in muscle are the E3 ubiquitin ligases muscle ring finger 1 (MuRF1) and muscle atrophy F-box (MAFbx; a.k.a. Atrogin-1). These genes, termed 'atrogenes,' are thought to control the majority of proteolysis in skeletal muscle across atrophic conditions (*Bodine et al., 2001*; *Gomes et al., 2001*). Knockout of MuRF1 results in a similar extent of muscle sparing

to NFκB inhibition in denervation (*Gomes et al., 2012*), unloading (*Labeit et al., 2010*), glucocorticoid treatment (*Baehr et al., 2011*), and inflammatory insult (*Files et al., 2012*), leading it, too, to be labeled 'essential' for skeletal muscle atrophy (*Peris-Moreno et al., 2020*). While compelling, this explanation remains incomplete. First, because inhibition of NFκB or MuRF1 signaling does not completely prevent atrophy induced by these various stimuli and second because the degree of prevention varies by stimulus and muscle. For example, muscle-specific *Ikbkb* knockout prevented ~70% of denervation-induced decrease in mass of the soleus muscle, but only ~30% in the gastrocnemius muscle (*Mourkioti et al., 2006*). Similarly, Iκβα super-repressor prevented only ~25% of the denervation-induced mass loss in the gastrocnemius and tibialis anterior muscles but ~50% of the tumor-induced loss in mass (*Cai et al., 2004*).

Recently, tendon tears (leading to release of muscle tension and muscle retraction) have emerged as potentially employing distinctive mechanisms of muscle atrophy. A comparative study of atrophic signaling between tenotomy and immobilization found that tenotomized gastrocnemius muscle atrophy was primarily characterized by a lysosomal/autophagic signature, in contrast to the classical ubiquitin-mediated proteasomal activity signature of the immobilized gastroc (*Bialek et al., 2011*). MuRF1 or MAFbx expression did not significantly increase with tenotomy, where dramatic increases were again shown with immobilization, denervation, and glucocorticoid-induced atrophy. This is somewhat surprising given the similarity in signaling induced by other models of unloading that remove either passive loading (e.g., hindlimb suspension and immobilization by casting) or active loading (e.g., denervation). In fact, previous to this study, tenotomy was assumed to employ the ubiquitin-mediated proteasome activity through NFκB and the atrogenes simply because of this similarity (*Laron et al., 2012*). More recent studies have focused on tenotomy of the rotator cuff (RC) muscles (without suprascapular nerve injury) as a model to investigate the mechanisms of muscle atrophy in human chronic RC tears. These studies have confirmed no increases in MuRF1 or MAFbx following RC tenotomy (*Liu et al., 2012*) and have uncovered additional evidence for increased autophagic flux (*Gumucio et al., 2012*; *Joshi et al., 2014*). Evidence in support of proteasomal activity in the early phase of RC tenotomy, however, also exists (*Valencia et al., 2017*), leaving open the possibility of coordinated action of multiple pathways.

Taken together, these prior studies suggest that although NFκB signaling is a critical component of many disparate atrophy models, it may not be ubiquitous – and in particular may not play a major role in tenotomy-induced atrophy. However, due to the complexity of NFκB signaling, its role in tenotomy-induced atrophy (or lack thereof) cannot be inferred from the lack of atrogene expression or the autophagic state of the muscle alone. First, because evidence suggests that NFκB is capable of driving atrophy independent of MuRF1 or MAFbx (*Cai et al., 2004*) and second, because there is a complex link between NFκB and autophagy which could cause NFκB to drive the autophagic process (reviewed in *Xia et al., 2021*). In this study, we sought to determine the role of NFκB signaling in tenotomy-induced atrophy of the RC muscles using muscle-specific gain- and loss-of-function IKKβ transgenic mouse models to inhibit and promote NFκB activity, respectively. We hypothesized that NFκB inhibition would ameliorate tenotomy-induced atrophy through an atrogene-independent mechanism, thus supporting NFκB inhibitors as a novel therapeutic strategy to maintain muscle mass following human chronic RC tears. However, we found that knockdown of IKKβ had no impact on the loss of muscle mass or contractile performance following tenotomy and that overexpression of constitutively active IKKβ was insufficient to drive muscle atrophy. Instead, our data point to an NFκB-independent, sex-specific mechanism driven by autophagy.

## Results

### Tenotomy-induced atrophy of the rotator cuff muscles is sex-specific

Change in muscle mass was the first outcome assessed following tenotomy of the supraspinatus (SS) and infraspinatus (IS) muscles – the primary abductors in the RC. Although this study was primarily concerned with differences between genotypes (due to changes in NFκB signaling), differences were noted between males and females in the wildtype group. Specifically, at week 8 (W8) post-tenotomy, mass loss in the male SS and IS muscles outpaced that in female mice (*Figure 1A*). Further investigation in the other outcomes of the study suggests that these sex-specific muscle mass effects arose from sex-specific differences in the morphological mechanisms of atrophy. Atrophy is driven by three

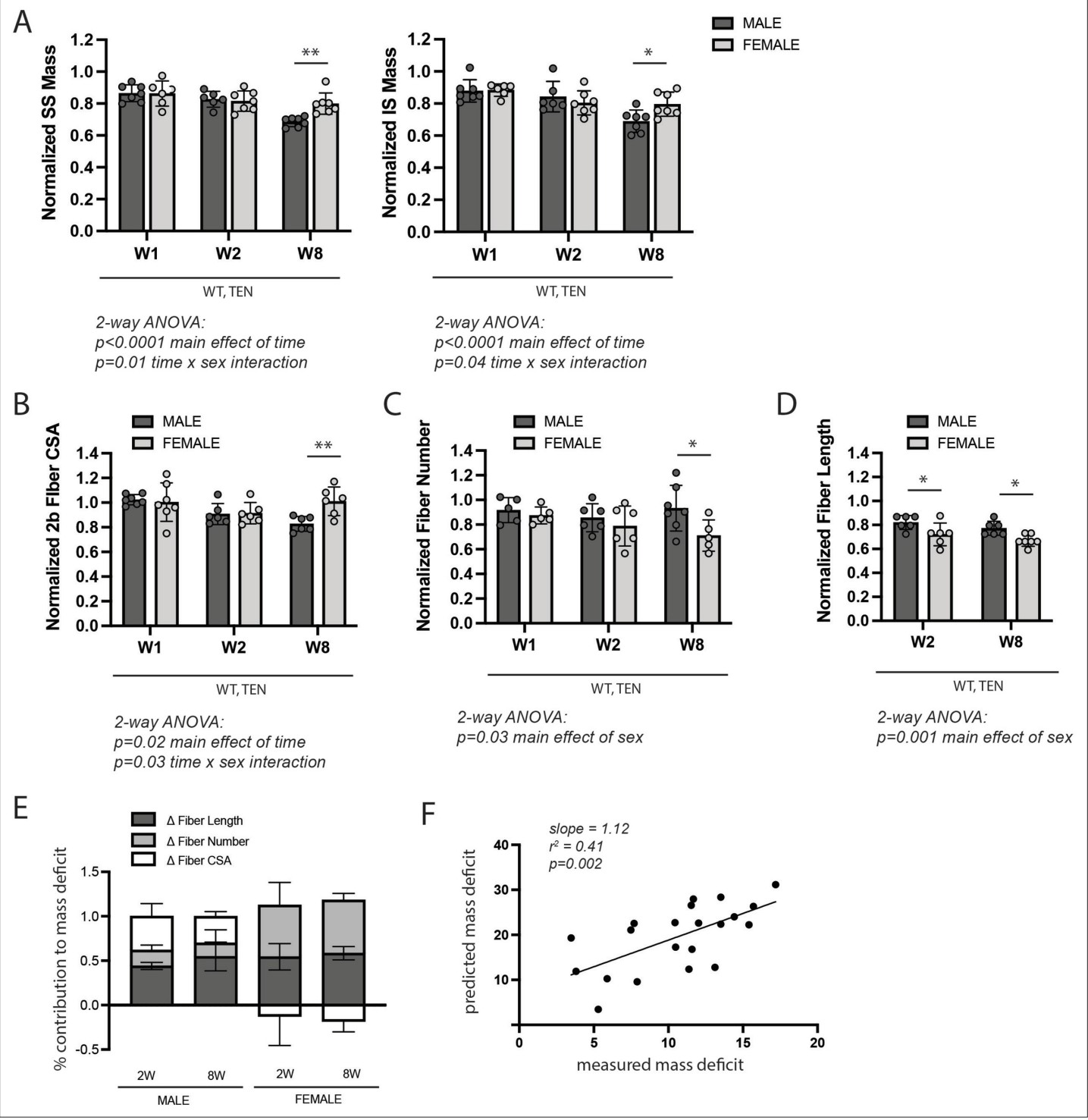

**Figure 1.** WT control mice exhibited sex-specific responses to late-stage tenotomy-induced muscle atrophy. (**A**) Supraspinatus (SS) and infraspinatus (IS) muscles from male mice lose more mass than female muscles at 8 weeks (W8) post-tenotomy (TEN). Muscle mass is normalized to body mass and the average value from the sham group for each sex. (**B**) Type 2b fiber cross-sectional area (CSA), assessed in histological sections, is also significantly reduced in males compared with females at W8. (**C**) The total number of fibers counted per histological section is significantly reduced in the female SS compared with male at W8. (**D**) Prediction of fiber lengths from measured muscle lengths during physiological testing indicates that female SS muscles lose a greater fraction of their fiber length at both W2 and W8 post-tenotomy. (**E**) Predictions of the contributions of each morphological change (**B–D**) to the measured mass deficit (**A**). Change in (Δ) fiber length (dark gray), fiber number (light gray), and fiber CSA (white) show different trends for males

*Figure 1 continued on next page*

*Figure 1 continued*

and females at W2 and W8 post-tenotomy. (**F**) The mass deficit *predicted* by morphological changes and the mass deficit *measured* at W2 and W8 are significantly correlated. (**A–D**) Raw values were normalized to the average of the sham group of the same sex. N = 5–7 per group; *p<0.05, **p<0.01.

The online version of this article includes the following source data for figure 1:

**Source data 1.** Raw data and statistical analysis results for normalized supraspinatus (SS) mass, normalized infraspinatus (IS) mass, normalized type 2b fiber cross-sectional area (CSA), normalized fiber number, normalized fiber length, % contribution to mass deficit, and predicted vs. measured mass deficit.

primary morphological adaptations: decreases in fiber sizes in the radial dimension (fiber atrophy), decreases in fiber numbers (hypoplasia), and decreases in fiber sizes in the longitudinal dimension (sarcomere subtraction). Only male mice demonstrated decreased fiber cross-sectional area (CSA) at W8 post- tenotomy (*Figure 1B*). Conversely, only female mice decreased fiber numbers at W8 post-tenotomy (*Figure 1C*). Female mice also decreased fiber lengths to a greater extent than male mice at both W2 and W8 (*Figure 1D*).

To better understand the relative impact of these morphological changes on muscle mass, a geometrical model was employed to predict the mass deficit from each change. Overall, changes in fiber length accounted for most of the mass deficit with tenotomy in both sexes (*Figure 1E*, purple). However, there were notable differences in the other sources. While changes in fiber CSA accounted for 25–30% of the mass loss in male mice at 2W and 8W, they accounted for none of the mass loss in female mice. In contrast, changes in fiber number account for 30–40% of the mass loss in females, but only 15–20% in males. To validate that our model captured the major sources of mass loss, we regressed the predicted mass deficit against the measured mass deficit for each muscle across both sexes. The measurements were significantly correlated ($r^2$ = 0.41, p=0.002; *Figure 1F*). Additionally, the slope of the regression line was close to 1, indicating that the morphological measurements were able to account for the majority of muscle mass loss following tenotomy. Given these differences, all subsequent analyses investigating the effect of IKKβ knockdown and overexpression on tenotomy-induced muscle atrophy consider sex-specific effects.

## IKKβ gain- and loss-of-function did not affect tenotomy-induced muscle atrophy

The gene encoding IKKβ (*Ikbkb*) was selectively deleted (IKKb^MKD) or constitutively activated (IKKb^MCA) in mature skeletal muscle fibers of 6–8-month-old transgenic mice. Tamoxifen treatment of IKKb^MKD mice caused a 50–60% reduction in the expression of *Ikbkb* mRNA in supraspinatus muscles that was sustained through W8 (*Figure 2A*). Conversely, tamoxifen treatment of IKKb^MCA mice caused a two- to threefold increase in *Ikbkb* expression at W8 (*Figure 2B*). Knockdown of *Ikbkb* expression decreased IKKβ protein levels by a comparable 50–60% (*Figure 1C*) while overexpression of constitutively active *Ikbkb* increased IKKβ protein levels by two- to threefold (*Figure 2D*). This deletion efficiency measured at the whole muscle level is comparable to other tamoxifen-inducible models in which satellite and interstitial cells also express the gene of interest. While the deletion efficiency in muscle fibers is likely higher than what is measured at the whole muscle level, it is likely incomplete and thus this model is referred to as knockdown rather than knockout. In male WT mice of the IKKb^MKD cohort, *Ikbkb* expression significantly increased by 40–50% in response to tenotomy at both W1 and W8 post-tenotomy (*Figure 2A*). However, this effect was not seen in the WT mice of the IKKb^MCA cohort (*Figure 2B*), nor was it evident at the protein level (*Figure 2C*). To ensure that a 50–60% reduction IKKβ protein was sufficient to impact NFκB signaling, we isolated the nuclear fraction from IKKb^MKD and WT muscle and quantified NFκB subunits p50 and p65, which reflect the NFκB-proteasomal degradation axis. Nuclear p50 and p65 were reduced by more than 50% in IKKb^MKD mice in both sham and tenotomy groups (*Figure 2E*), confirming inhibition of NFκB nuclear translocation. Furthermore, this assay was validated using positive controls (*Figure 2—figure supplement 1*). Thus, these genetic manipulations resulted in a consistent two- to three3fold knockdown or overexpression of IKKβ in both sexes.

In both sham and tenotomy groups, IKKβ conditional deletion caused a small increase in mass of the SS (*Figure 3A*) and IS (*Figure 3B*) muscles normalized to body mass. Three-way ANOVA applied within each treatment group showed a significant main effect of genotype in SS sham, SS tenotomy, and IS sham comparisons. However, only in the male SS at W1 post-tenotomy was the individual

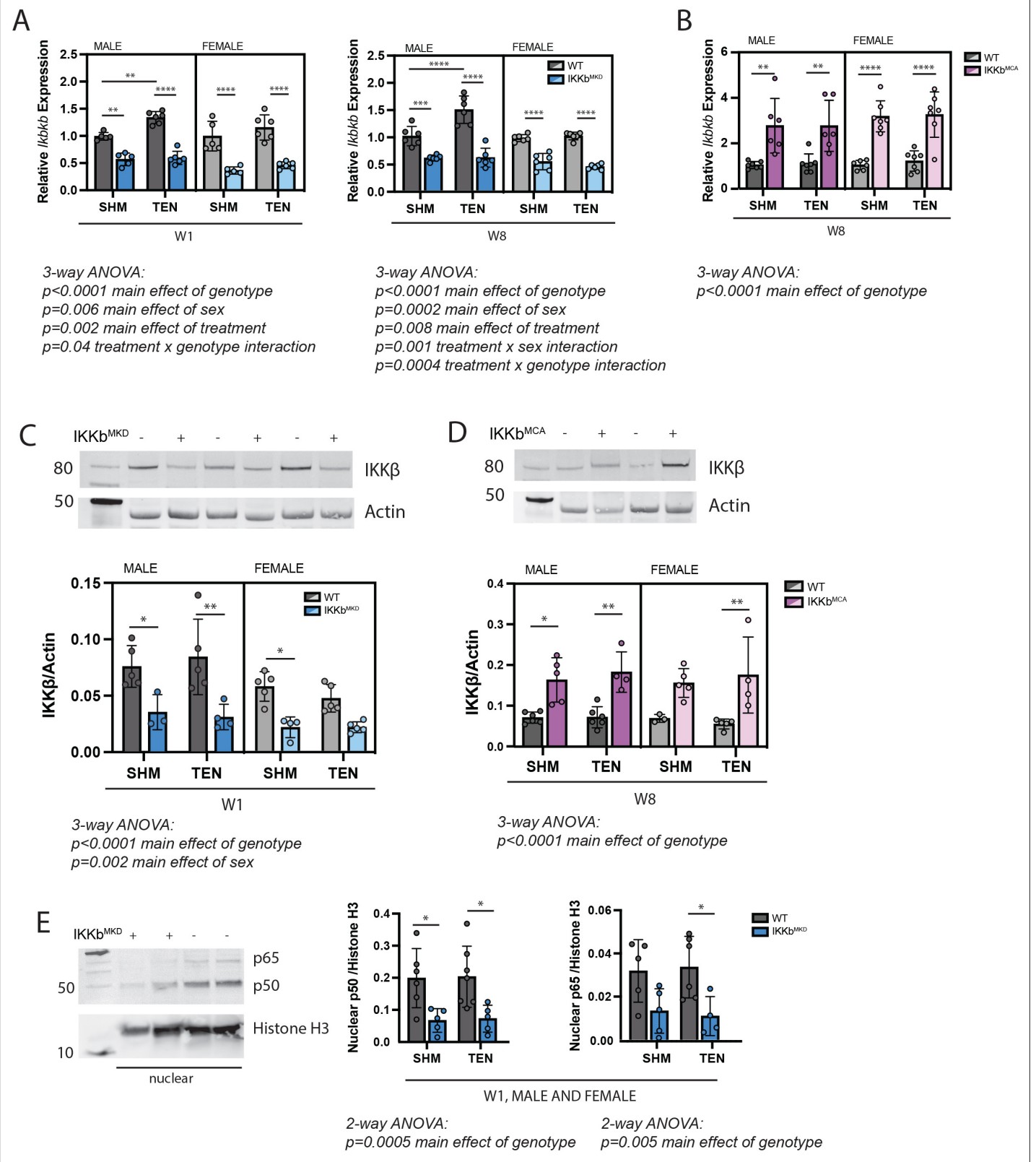

**Figure 2.** Muscle-specific inducible deletion and overexpression of IKKβ induced a twofold decrease and increase in IKKβ expression, respectively. (**A**) In mice with muscle-specific inducible IKKβ deletion (IKKb^MKD), expression of the IKKβ gene (*Ikbkb*) was reduced 50–60% in male and female sham (SHM) and tenotomized (TEN) supraspinatus (SS) at week 1 (W1) and week 8 (W8) compared with wildtype (WT). (**B**) In mice with muscle-specific inducible overexpression of constitutively active IKKβ (IKKb^MCA), *Ikbkb* expression was increased two- to threefold in SHM and TEN groups at W8. (**C**) Protein

*Figure 2 continued on next page*

*Figure 2 continued*

abundance of IKKβ was 50–60% lower in IKKb^MKD compared with WT. (**D**) Protein abundance of IKKβ was two- to threefold higher in IKKb^MCA compared with WT. (**E**) Protein abundance of NF $\kappa$ B subunits p50 and p65 in the nuclear fraction of SS muscles was reduced 50–60% in IKKb^MKD compared with WT. N = 3–6 per group; *p<0.05, **p<0.01, ***p<0.005, ****p<0.001.

The online version of this article includes the following source data and figure supplement(s) for figure 2:

**Source data 1.** Raw data and statistical analysis results for relative *Ikkb* expression, IKKβ/actin, and nuclear p50/histone H3.

**Source data 2.** Western blot raw and uncropped images.

**Figure supplement 1.** Validation of assays with positive controls.

**Figure supplement 1—source data 1.** Nuclear p50/histone-H3.

**Figure supplement 1—source data 2.** Western blot raw and uncropped images.

between-genotype comparison significant. To determine whether IKKβ conditional deletion uniquely affected the atrophic process post-tenotomy or simply increased muscle mass generally, tenotomized SS and IS muscle masses were normalized to the mass of the intact tibialis anterior (TA) of the same mouse. In this normalization scheme, no consistent difference was observed between genotypes in either muscle, either sex, or any timepoint. Conversely, overexpression of caIKKβ exacerbated tenotomy-induced atrophy to a very minor extent in both SS and IS muscles (*Figure 3D and E*), which resulted in a significant treatment × genotype interaction effect by three-way ANOVA. Normalizing to the mass of the TA eliminated these effects (*Figure 3F*). Tenotomized muscles in IKKb^MKD mice exhibited the same sex specificity in mass loss (*Figure 3A and B*) as noted for their WT littermates (*Figure 1*). Tenotomized muscle mass from the IKKb^MCA cohort did not show this effect when normalized to body mass, but there was a significant difference in body mass between sham and tenotomized male mice that may have obscured the atrophic sex specificity. Both SS and IS masses normalized to TA mass showed significant main effects of sex by two-way ANOVA (*Figure 3F*).

## IKKβ gain- and loss-of-function did not affect tenotomy-induced muscle function loss

Following tenotomy, SS and IS muscles lose ~20–30% of their specific peak tetanic contractile tension (*Figure 4*; *Meyer, 2022*). This loss was not affected by IKKβ deletion or overexpression of caIKKβ in either muscle or either sex at any timepoint. Unlike mass loss, contractile deficits were similar between W2 and W8 in both sexes. This suggests that normalizing contractile tension to physiological cross-sectional area (PCSA) fully accounts for the ongoing mass loss.

## Morphological adaptations following tenotomy were independent of IKKβ knockdown

We next sought to explore whether the cellular underpinnings of the tenotomy-induced muscle mass loss and contractile dysfunction were modified by IKKβ knockdown. For these assays, we focused solely on the SS muscle since SS and IS responded similarly to tenotomy with regard to mass and contractile outcomes. Surprisingly, tenotomy caused relatively small changes in fiber CSA compared to other forms of atrophy for a similar mass loss. At W8 (the timepoint of maximal mass deficit), only type 2b fibers (*Figure 5A*, green) from male muscles exhibited a leftward shift in the histogram of CSA with tenotomy (*Figure 5B*, green), while type 2a fibers exhibited a rightward shift (*Figure 5B*, blue). Analysis of mean CSA of type 2a and type 2b fibers over time further illustrated this effect. There was a main effect of time in analysis of type 2a CSA by three-way ANOVA owing to the continued increase in CSA through W8 (*Figure 5C*). There was also a main effect of sex, with a significant sex x time interaction effect in type 2b CSA (*Figure 5D*) owing to the sex specificity presented in *Figure 1*. The mass loss unaccounted for by fiber CSA changes was due to decreases in fiber numbers and lengths (*Figure 5E and F*). Fiber length had an additional main effect of sex (*Figure 5F*) as the sex specificity presented in *Figure 1* was evident in the IKKb^MKD group as well. There were no significant main effects or interactions involving genotype by three-way ANOVA in any of the comparisons, indicating that IKKβ knockdown did not shift the mechanisms of mass loss. In contrast to morphological changes, three-way ANOVA of fiber-type distributions showed no consistent shifts in any fiber-type percentage as a function of tenotomy or IKKβ knockdown (*Figure 5—figure supplement 1*).

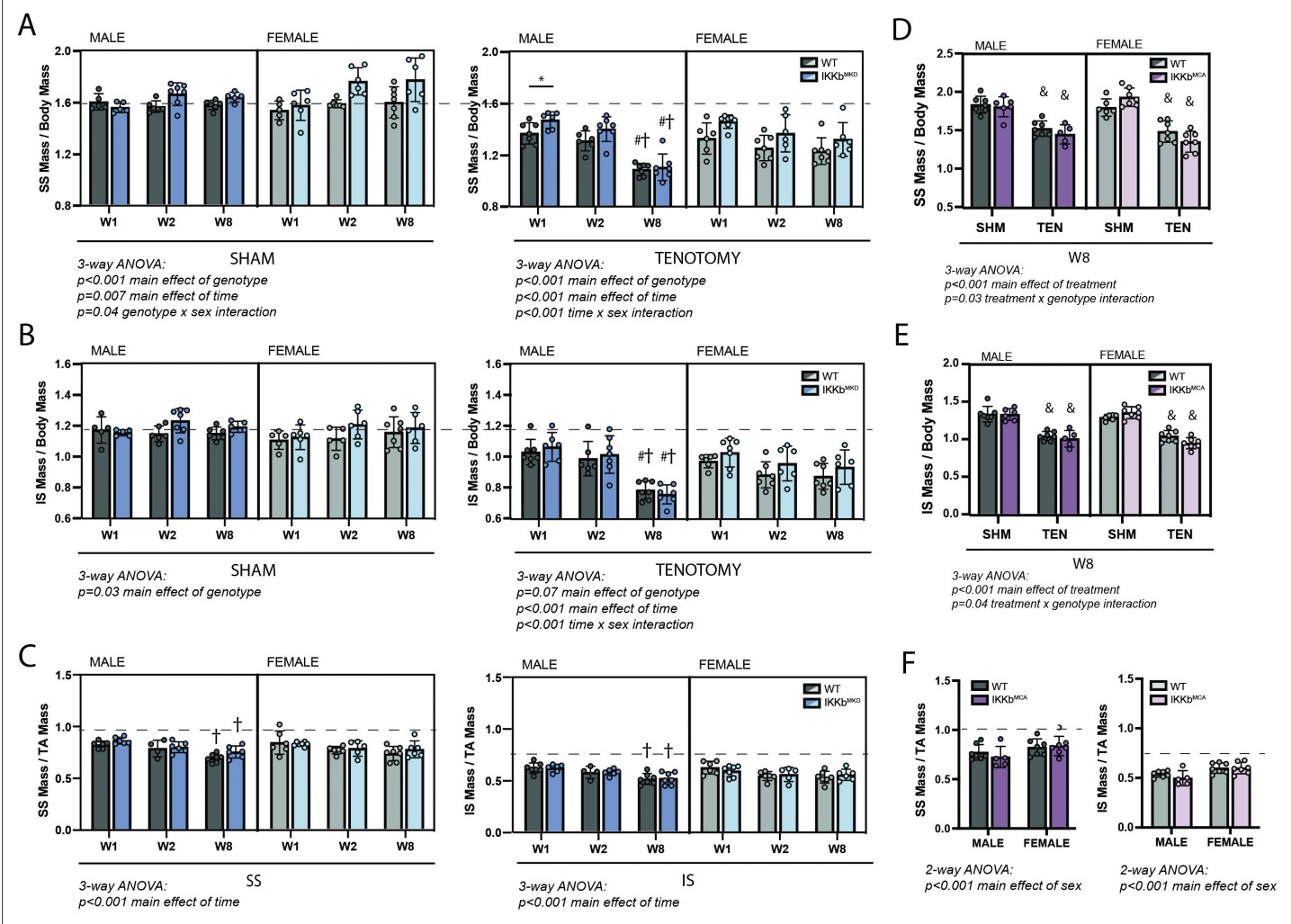

**Figure 3.** IKKβ deletion and overexpression modestly affected muscle mass in sham (SHM) and tenotomy (TEN) groups without uniquely affecting muscle loss post-tenotomy. (**A**) Supraspinatus (SS) muscles from IKKb^MKD mice had a minor increase in mass normalized to body mass in both SHM (left) and TEN (right) groups compared with WT. (**B**) Infraspinatus (IS) muscles from IKKb^MKD mice similarly had a minor increase in normalized mass compared with WT. (**C**) When SS mass was normalized to tibialis anterior (TA) mass, there was no difference between IKKb^MKD and WT groups. (**D**) IS muscles from IKKb^MCA mice had a minor decrease in muscle mass in the TEN group at week 8 (W8) compared with WT. (**E**) IS muscles from IKKb^MCA mice similarly had a minor decreased in normalized mass compared with WT in the TEN group only. (**F**) When SS mass was normalized to TA mass, there was no difference between IKKb^MCA and WT groups. (**A–F**) Dotted lines are the average of all sham values included for reference. *p<0.05 between genotypes within the same sex and timepoint. †p<0.05 compared with week (W1) values within the same sex and genotype. N = 5–7 per group; &p<0.05 compared with sham values within the same sex and genotype.

The online version of this article includes the following source data for figure 3:

**Source data 1.** Raw data and statistical analysis results for normalized supraspinatus (SS) mass/body mass, infraspinatus (IS) mass/body mass, SS mass/tibialis anterior (TA) mass and IS mass/TA mass.

## Tenotomy-induced structural pathology points to sex specificity in autophagy

Next, we assessed metrics of structural pathology that are associated with tenotomy and thought to impact muscle contraction – fibrosis, fatty infiltration, and fiber degeneration – and related progenitor populations. Fibrosis and fatty infiltration are progressive in tenotomy (*Rubino et al., 2007*), and were therefore measured at W8 to capture the highest signal (*Figure 6A*). Fibrosis, assessed as the area fraction positive for Sirius Red, was elevated following tenotomy but without a main effect of genotype or sex, indicating that tenotomy induced fibrosis equally in WT and IKKβ knockdown males and females (*Figure 6B*). Fatty infiltration, assessed as the area fraction positive for Oil Red-O, was

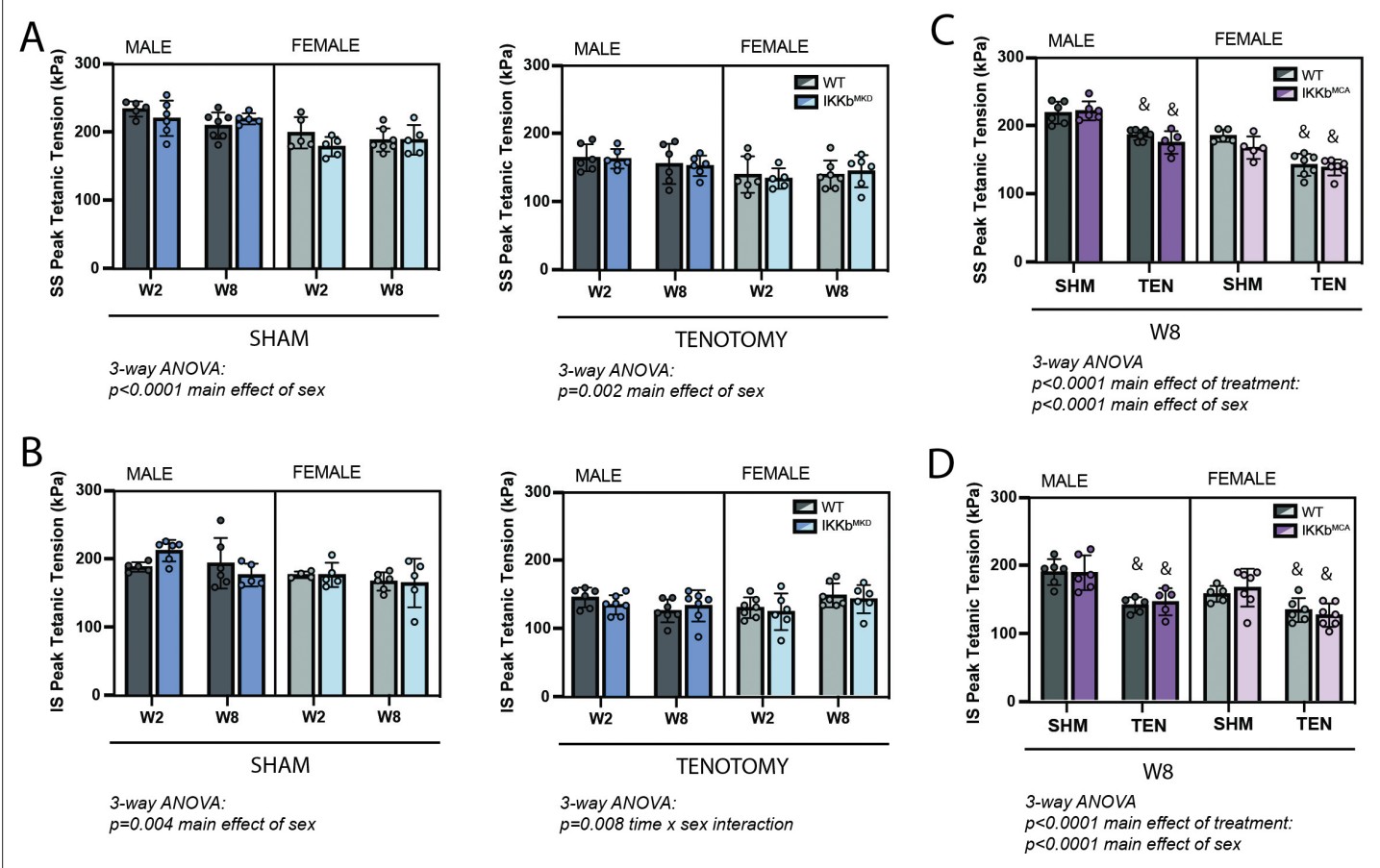

**Figure 4.** IKKβ deletion and overexpression did not affect the loss of tetanic tension following tenotomy (TEN). (**A**) Peak tetanic tension of supraspinatus (SS) muscles normalized to physiological cross-sectional area (PCSA) is not different between WT and IKKb[MKD] genotypes in sham (SHM) (left) and TEN (right) groups. (**B**) Peak tetanic tension of infraspinatus (IS) muscles is also not different between WT and IKKb[MKD] genotypes in SHM (left) and TEN (right) groups. (**C**) Peak tetanic tension of SS muscles is not different between WT and IKKb[MCA] genotypes in SHM and TEN groups. (**D**) Peak tetanic tension of IS muscles is not different between WT and IKKb[MCA] genotypes in SHM and TEN groups. N = 5–7 per group; &p<0.05 compared with sham values within the same sex and genotype.

The online version of this article includes the following source data for figure 4:

**Source data 1.** Raw data and statistical analysis results for supraspinatus (SS) and infraspinatus (IS) peak tetanic tension.

also elevated following tenotomy, an effect that was exacerbated in female mice due to the overall higher levels of intramuscular fat in females (***Figure 6C***; ***McHale et al., 2012***). However, there was no main effect of genotype, again indicating that IKKβ knockdown did not affect the progression of fatty infiltration. Fiber degeneration was assessed on hematoxylin & eosin (H&E)-stained sections by a blinded rater counting necrotic fibers (active degeneration) and centrally nucleated fibers (regeneration) (***Figure 6D***). Across timepoints, there were no obvious necrotic fibers in tenotomized muscle, suggesting that active degeneration was either not occurring or missed at the timepoints chosen for assessment. However, there was a gradual increase in centrally nucleated fibers over time following tenotomy, which was exacerbated in males (***Figure 6E***). A blinded rater noted the appearance of basophilic puncta (hematoxylin-positive, DAPI-negative structures) in some H&E-stained sections (***Figure 6D and F***; single arrows). These were generally more prevalent in males than females, and this difference was exacerbated at 8W, where they were increased 20-fold over W1 levels (***Figure 6G***). These puncta stained positive for LC3 and p62, suggesting that they were autophagic vesicles. In line with the increase in fibrosis and fatty infiltration seen at W8, the population of fibro-/adipogenic progenitors (FAPs) identified by immunostaining for PDGFRα was increased at week 1 post-tenotomy (***Figure 6H and I***) in both sexes. In contrast, the population of satellite cells identified by nuclear Pax7 did not change with tenotomy at week 1, suggesting minimal regenerative response in the muscle

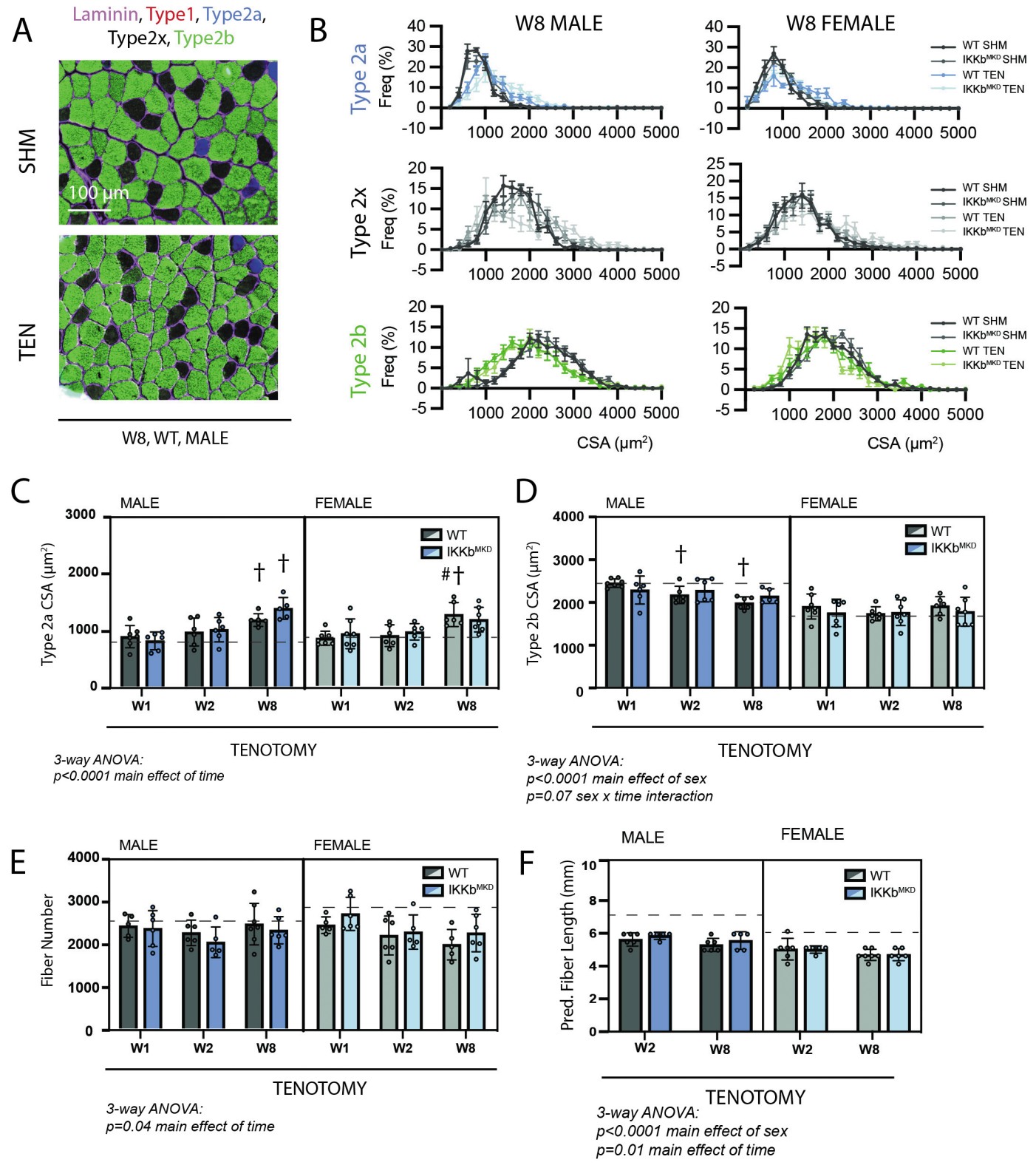

**Figure 5.** Muscle morphological changes following tenotomy (TEN) were not affected by IKKβ knockdown and were sex-specific. (**A**) Representative fiber-type staining of histological sections from sham (SHM) and TEN WT supraspinatus (SS) muscles at week 8 (W8). Fiber types were identified by immunostaining against isoforms of myosin heavy chain – type 1 (red), type 2a (blue), type 2x (black), and type 2b (green) – and area was quantified by the laminin border (magenta). (**B**) Distribution of cross-sectional areas (CSA) for type 2a, 2x, and 2b fibers for male and female mice at W8. Histograms

*Figure 5 continued on next page*

*Figure 5 continued*

of type 2b fiber CSA in male mice are shifted to the left following TEN. (**C**) Average CSA of type 2a fibers and (**D**) average CSA of type 2b fibers across genotypes and timepoints within the TEN group. Only type 2b fiber CSA in males decreases following tenotomy. (**E**) Count of the total number of fibers in the SS cross-section across genotypes and timepoints within the TEN group. (**F**) Prediction of fiber length from measurements of muscle length during physiological testing at W2 and W8 within the TEN group. No data exist at week 1 because physiological testing was not performed at that timepoint. (**C–F**) The dotted lines are the average of all sham values included for reference for each sex. N = 5–7 per group; †p<0.05 compared with week (W1) values within the same sex and genotype.

The online version of this article includes the following source data and figure supplement(s) for figure 5:

**Source data 1.** Raw data and statistical analysis results for fiber cross-sectional area (CSA) histograms, fiber type CSA, fiber number, and predicted fiber length.

**Figure supplement 1.** Fiber-type distribution is not altered by tenotomy or IKKβ knockdown.

**Figure supplement 1—source data 1.** Raw data and statistical analysis results for fiber-type distributions.

following tendon transection. This aligns with the modest increase in central nuclei that account for less than 3% of fibers even at W8.

## Knockdown of IKKβ did not induce the typical atrophic signaling pathways after tenotomy

We investigated which atrophic signaling pathways were active during the early and late phases of tenotomy responses. The ubiquitin-proteasome pathway may be activated during the early phase of tenotomy, consistent with a previous report (*Valencia et al., 2017*). While the typical atrogenes MuRF1 and MAFbx were not substantially elevated at W1, expression of Foxo3, an additional driver of both proteolysis and autophagy (*Zhao et al., 2007*), was significantly increased at W1 in female mice (*Figure 7A*). Expression of all three genes decreased at W8, suggesting that they are not driving late-stage atrophy. Consistent with this, protein ubiquitination, assessed by Western blot against ubiquitin, was modestly increased at W1 but not at W8 post-tenotomy (*Figure 7B*). To examine autophagy, we assessed gene expression and protein abundance of components of the autophagasome. While Becn1 and Bnip3 increased expression and Atg5 decreased expression significantly from W1 to W8, these changes were very minor (*Figure 7C*). Other components remained unchanged. Similarly, there were no differences in the LC3II/LC3I ratio or p62 at W1, but a modest increase at W8 with tenotomy, where three-way ANOVA showed a significant main effect of treatment (*Figure 7D*). Interestingly, three-way ANOVA of Gaparapl1, Becn1, and Bnip3 expression as well as protein levels for p62 found a main effect of sex owing to higher levels in male mice than female further supporting a sex specificity in autophagic processing. Overall, neither the ubiquitin-proteolysis nor autophagy-lysosome pathway were substantially affected by tenotomy at W1 or W8. Furthermore, knockdown of IKKβ did not impact these small effects. As the loss of muscle mass could be driven by decreased protein synthesis instead of increased protein degradation, we also assessed activation of the Akt/mTOR pathway, which can mediate increased protein synthesis in response to loading changes. We did not find any changes in phosphorylation of Akt, mTOR, or S6 ribosomal protein with tenotomy or IKKβ knockdown (*Figure 7—figure supplement 1*). Finally, there were no differences in myostatin expression, suggesting that tenotomy-induced atrophy is also not likely driven by myostatin (*Figure 7—figure supplement 1*).

## Discussion

This study investigated the effects of NFκB inhibition on muscle atrophy following tenotomy. Using a transgenic muscle-specific and inducible deletion of IKKβ, we were able to block more than 50% of the nuclear translocation of NFκB subunits in muscle fibers at the time of tenotomy of the RC muscles. While this caused a mild increase in muscle mass in general, it surprisingly had no additional effect on any outcomes following tenotomy, including muscle mass and contractile force, fiber area, length and number, and atrophic signaling through the ubiquitin-proteasome and autophagy pathways. Similarly, doubling IKKβ expression using a gain-of-function mouse model had little effect on muscle responses to tenotomy. Together, these results indicate that NFκB does not play a major role in mediating tenotomy-induced atrophy. Interestingly, this study also uncovered sex-specific mechanisms of

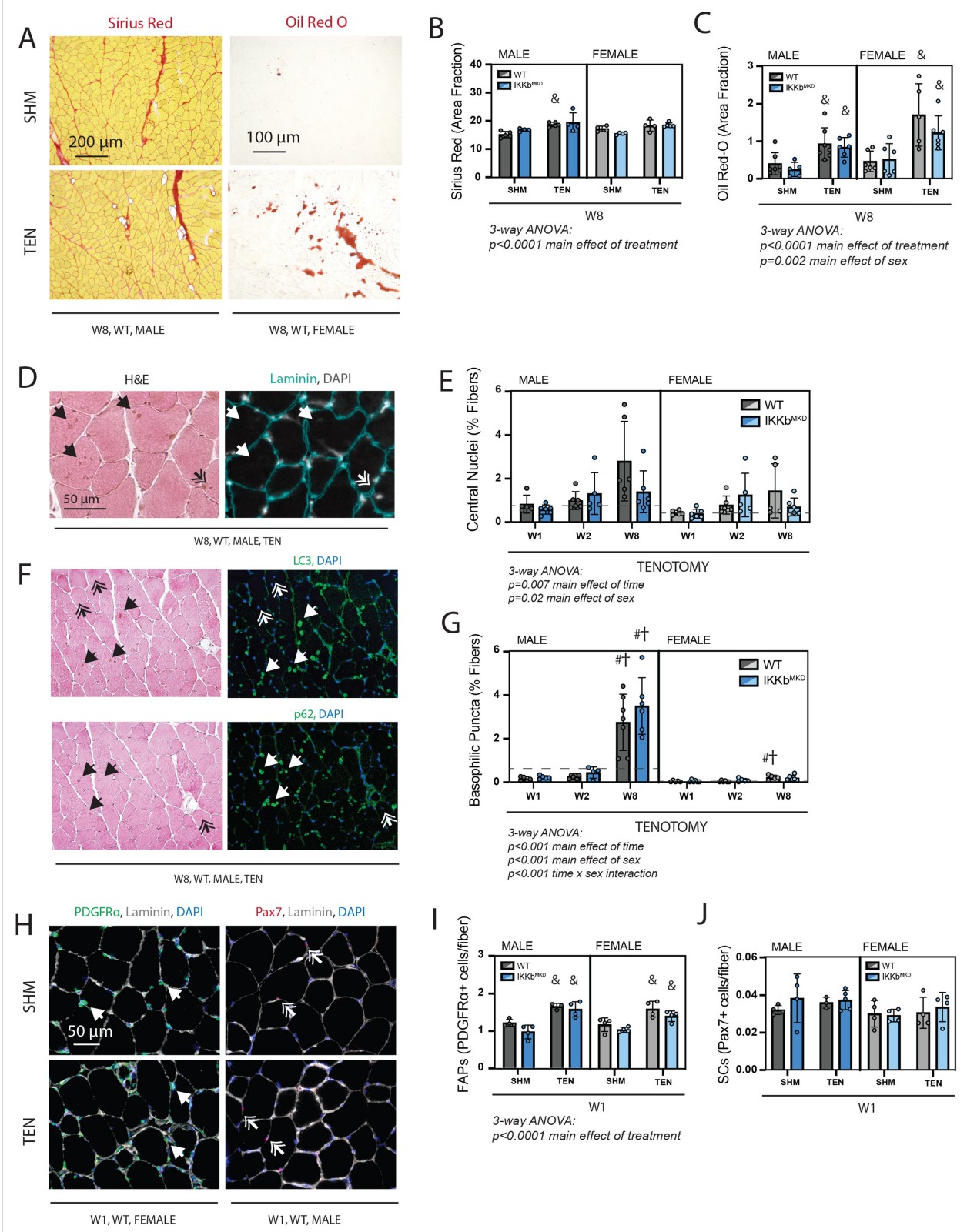

**Figure 6.** Cellular pathology following tenotomy (TEN) was not affected by IKKβ knockdown and suggests sex specificity in autophagic flux. (**A**) Representative Sirius Red and Oil Red-O staining of histological sections from sham (SHM) and TEN WT supraspinatus (SS) muscles at week 8 (W8). The red fraction in Sirius Red indicates concentrations of collagen, while the red fraction in Oil indicates intramuscular adipocytes. (**B**) Quantification of the fraction of the section area occupied by Sirius Red as a marker of fibrosis for male and female mice at W8 shows moderate increases with TEN

*Figure 6 continued on next page*

*Figure 6 continued*

in both genotypes. (**C**) Quantification of the fraction of section area occupied by Oil Red O as a marker of fatty infiltration for male and female mice at W8 shows significant increases with TEN in both genotypes. (**D**) Representative sections stained with H&E and laminin with DAPI used to identify centralized nuclei (double arrows) as hematoxylin and DAPI-positive structures central within the laminin boundary and other basophilic puncta (arrows) as hematoxylin-positive central structures negative for DAPI. (**E**) Quantification of fibers with centralized nuclei as a percentage of all fibers as a marker of regeneration across genotypes and timepoints within the TEN group. (**F**) Representative sections sequentially stained with LC3 or p62 followed by H&E to identify LC3/p62 and hematoxylin positivity in the same section. (**G**) Quantification of fibers with basophilic puncta as a percentage of all fibers across genotypes and timepoints within the TEN group. Basophilic puncta were more abundant in male SS sections at W8. (**D, F**) Dotted lines are the average of all sham values included for reference for each sex. N = 5–7 per group. (**H**) Representative immunostaining for PDGFRα (green) and Pax7 (red) identifying fibro-/adipogenic progenitors (FAPs) and satellite cells (SCs), respectively. (**I**) Quantification of FAPs shows an increase in tenotomized groups across genotypes and sexes. (**J**) Quantification of SCs shows no change with TEN. †p<0.05 compared with week (W1), #p<0.05 compared with W2, and &p<0.05 compared with sham values within the same sex and genotype.

The online version of this article includes the following source data for figure 6:

**Source data 1.** Raw data and statistical analysis results for Sirius Red (area fractions), Oil Red-O (area fractions), central nuclei (%fibers), basophilic puncta (% fibers), fibro-/adipogenic progenitors (FAPs) (PDGFRα + cells/fiber), and satellite cells (SCs) (α7 integrin + cells/fiber).

tenotomy-induced atrophy, most evident at the later timepoints investigated. Male, but not female, mice continued to lose muscle mass between 2W and 8W post-tenotomy. The sex-specific result was likely due to ongoing type 2b fiber atrophy driven by accumulation of autophagic vesicles, which was exacerbated in males at 8W. This surprising finding warrants further investigation in naïve wildtype mice.

The finding that IKKβ knockdown had little effect on tenotomy-induced muscle atrophy aligns with the limited data in tenotomy models, but is contrary to the NFκB muscle atrophy dogma. Transgenic interference with NFκB signaling substantially reduced loss of muscle mass across nearly every model investigated, including unloading, immobilization, denervation, acute inflammatory challenge, nutrient deprivation, and cancer (*Cai et al., 2004*; *Mourkioti et al., 2006*; *Judge et al., 2007*; *Van Gammeren et al., 2009*; *Reed et al., 2011*; *Haegens et al., 2012*; *Langen et al., 2012*; *Lee and Goldberg, 2015*). These models encompass diverse stimuli for atrophy that are frequently subcategorized as disuse, aging, or disease (*Romanick et al., 2013*). Tenotomy is typically included in the disuse category, along with unloading, immobilization, and denervation. However, tenotomy is fundamentally unique in that it severs the connection of the muscle to the skeleton, eliminating passive tension and immediately changing the length of the muscle as the muscle recoils and comes to rest in a shortened position. Importantly, tenotomy alone does not involve any direct damage to the muscle or its innervation, avoiding confounding effects of denervation and extensive regeneration. While RC tenotomy is frequently paired with denervation to elicit more dramatic muscle pathology (particularly fatty infiltration), there is not strong evidence that nerve damage is a consistent feature of even massive RC tears in humans (*Costouros et al., 2007*). Similarly, extensive regeneration does not appear to be a notable feature of RC tears in humans, where only ~10% of fibers exhibit central nucleation, compared with ~3% in controls (*Gibbons et al., 2017*). In the mouse, we find no proliferation of satellite cells at W1 following tenotomy and only a minor (<3%) increase in centrally nucleated fibers at later timepoints. This, combined with the lack of dilution of the NFkb knockdown at the gene and protein level, suggests that the genetically unmodified SCs are not contributing substantial additional myonuclei. However, unlike human tears, rodent models of tenotomy develop a fibrous 'pseudo-tendon' scar in the gap between the tendon and bone, partially restoring passive loading to the muscle (*Jamali et al., 2000*). This retraction plus scarring in the shortened position induces the loss of muscle mass in the longitudinal dimension by sarcomere subtraction (*Ward et al., 2010*; *Meyer, 2022*); a phenomenon that does not occur in hindlimb unloading or denervation models and, to a lesser extent, in immobilization models (*Heslinga and Huijing, 1993*; *Van Dyke et al., 2012*). Tenotomy and immobilization are also the only disuse models to demonstrate central core lesions of myofibril breakdown, hypothesized to be due to contraction in the shortened position. However, these are also exacerbated in tenotomy compared with immobilization (*Baewer et al., 2004*; *Baewer et al., 2008*). This suggests that tenotomized muscle is engaged in high levels of active remodeling, differentiating it from models reducing excess muscle mass in response to decreased use. This active remodeling could be responsible for the unique degradation signature in tenotomy compared with immobilization or denervation (*Bialek et al., 2011*; *Liu et al., 2012*; *Joshi et al., 2014*), where either the myofibril breakdown or

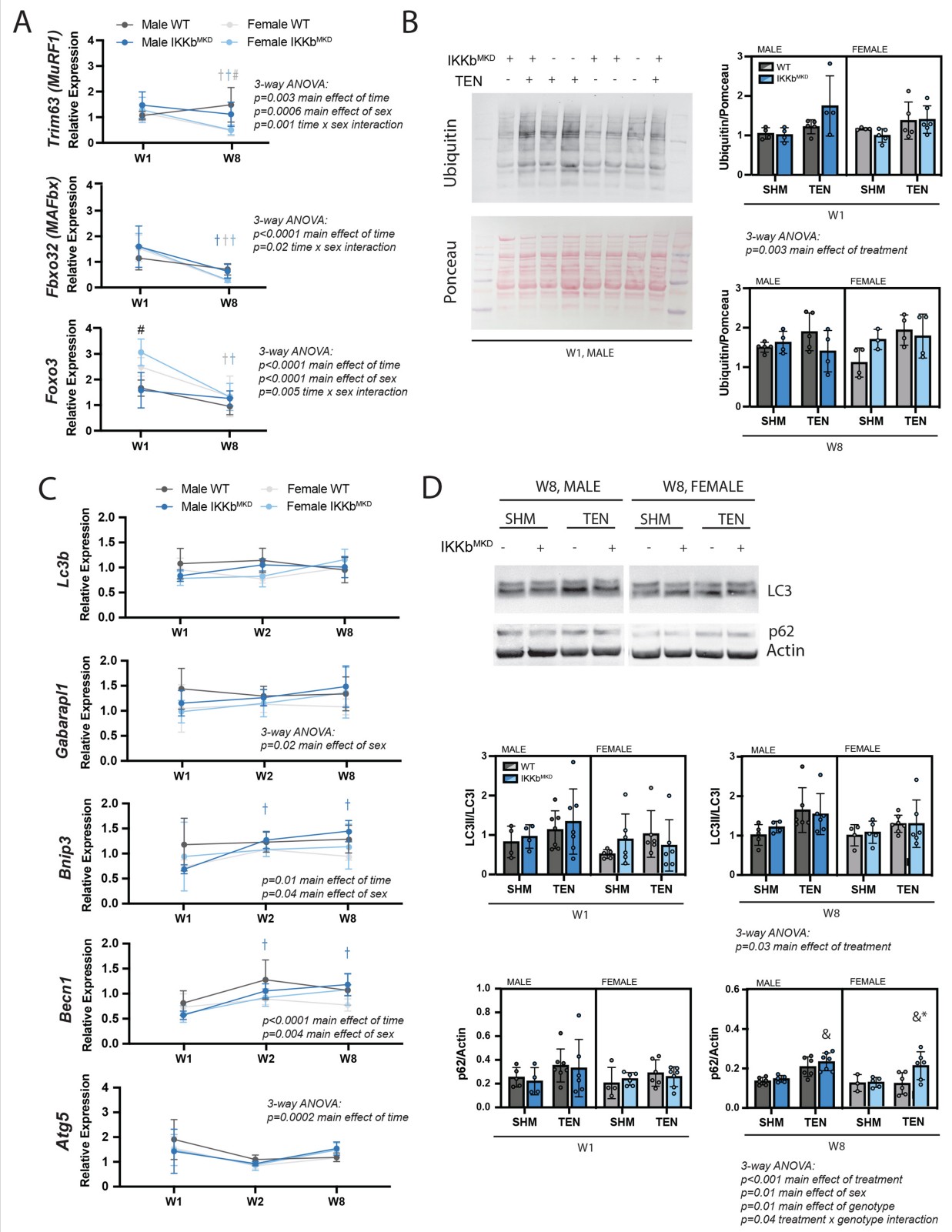

**Figure 7.** IKKβ knockdown did not impact the major signaling markers of the ubiquitin-proteasome or autophagy-lysosome pathways. (**A**) Quantification of MuRF1, MAFbx, and Foxo3 gene expression normalized to GAPDH and week 1 (W1) WT male sham in male and female mice at W1 and week 8 (W8). (**B**) Quantification of the abundance of ubiquitin protein normalized to total protein stained with Ponceau at W1 and W8. (**C**) Quantification of Lc3b, Gabarapl1, Bnip3, Becn1, and Atg5 gene expression normalized to GAPDH and W1 WT male sham at W1 and W8. (**D**) Quantification of the ratio of

*Figure 7 continued on next page*

*Figure 7 continued*

LC3II to LC3I and p62 protein abundance at W1 and W8. N = 4–6 per group; #p<0.05 compared with male values within the same genotype. †p<0.05 compared with W1. &p<0.05 compared with sham values within the same sex and genotype, *p<0.05 compared with WT values within the same sex and treatment.

The online version of this article includes the following source data and figure supplement(s) for figure 7:

**Source data 1.** Raw data and statistical analysis results for Trim63, Fbxo32, and Foxo3 gene expression, ubiquitin/Ponceau, Lc3b, Gabarapl1, Bnip3, Becn1, and Atg5 expression, LC3II/LC3I, and p62/actin.

**Source data 2.** Western blot raw and uncropped images.

**Figure supplement 1.** Akt/mTOR/S6 ribosomal phosphorylation is unchanged in response to tenotomy or IKKβ knockdown.

**Figure supplement 1—source data 1.** Raw data and statistical analysis results for pAkt/Akt, pmTOR/mTOR, pS6/S6, and relative Mstn gene expression.

**Figure supplement 1—source data 2.** Western blot raw and uncropped images.

**Figure supplement 2.** Validation of assays with positive controls.

**Figure supplement 2—source data 1.** Raw data and statistical analysis results for relative MuRF1 and MAFbx gene expression, ubiquitin/Ponceau, and LC3II/LC3I.

**Figure supplement 2—source data 2.** Western blot raw and uncropped images.

selective serial sarcomere subtraction is engaging autophagic mechanisms. Our data also suggest that the mechanisms of muscle loss may vary temporally following tenotomy. IKKβ knockdown had the greatest effect on tenotomized muscle mass at W1 (*Figure 3A*), which aligned with main the effects of treatment in ubiquitin protein abundance and MuRF1 expression (*Figure 7A and B*). These transient effects could be due to a differential atrophic responses of the muscle pre- and post-scarring or due to the resolution of inflammatory signals following tenotomy.

Interestingly, while male mice continued to lose muscle mass between W2 and W8 post-tenotomy, the contractile deficit stabilized (*Figure 3*). Peak tetanic tension is normalized to PCSA, which theoretically accounts for changes in fiber CSA/number and serial sarcomere subtraction, but even after this normalization, a deficit remained. The source of this deficit is unknown. It could be that PCSA is not fully accounting for the impact of morphological changes on contractile force or the deficit may be attributed to another factor, such as muscle quality. The fact that contractile properties remained stable during ongoing atrophy in male mice suggests that PCSA does sufficiently account for these changes. Additionally, data in the tenotomized rat SS indicate that contractile deficits precede morphological changes and are measurable as soon as 1 day post-tenotomy, remaining stable for 2 weeks (*Valencia et al., 2017*). This suggests that the contractile deficit originated early in the tenotomy injury and never resolved during the subsequent process of healing. There is considerable evidence for myofibrillar disruption and fiber damage post-tenotomy (reviewed in *Jamali et al., 2000*), and it is possible that the tenotomized muscle cannot adequately repair this damage in its unloaded state. If this is the case, it also offers an explanation for why IKKβ knockdown did not affect the contractile deficits. However, it is difficult to say whether some recovery of this initial damage occurs and is offset by the later development of structural pathology (e.g., fibrosis, fat infiltration, basophilic puncta, etc.) or whether the degree of this later structural pathology is insufficient to drive additional measurable contractile changes, as other work suggests it does (*Biltz et al., 2020*). Future studies using targeted mouse models such as the lipodystrophic mouse (that does not develop fat infiltration) or pharmacologics that target accumulation of autophagic vesicles could shed some light on this question.

Expression of a constitutively active form of IKKβ doubled the concentration of IKKβ in mice, yet this increase had negligible effects on tenotomy-induced atrophy. This finding is in contrast to findings from the MIKK mouse and transfection of constitutively active IKKβ, which reportedly induced significant muscle atrophy, even in the absence of an unloading stimulus (*Cai et al., 2004*; *Van Gammeren et al., 2009*). However, these studies notably induced more than tenfold increases in constitutively active IKKβ, levels that may be supra-physiological. Our findings suggest that smaller increases in IKKβ may not be sufficient to drive atrophy without an additional stimulus.

Increased protein ubiquitination has been observed early (2–15 days) post-tenotomy (*Baewer et al., 2008*; *Valencia et al., 2017*), which could indicate coordinated action of multiple activated pathways, one of which may be regulated by NFκB. Due to this mounting evidence for autophagy due to tenotomy, it was reasonable to hypothesize in this study that NFκB might also be involved in the

atrophic process. Additionally, NFκB could directly affect autophagy as several autophagic regulatory genes (Beclin 1, Atg6, and LC3) are targets of NFκB in other tissues (*Copetti et al., 2009*; *Nivon et al., 2009*). While our data strongly suggest that NFκB does *not* play a major role in tenotomy-induced atrophy, there are several limitations that could have masked some NFκB-driven effects. First, HSA-driven deletion of *Ikbkb* was only 50–60% efficient, leaving 40–50% of the endogenous levels of IKKβ to participate in NFκB signaling. Indeed, we found 40–50% of endogenous WT NFκB subunits p50 and p65 present in the nuclear fraction. By comparison, constitutive deletion of *Ikbkb* under control of muscle creatine kinase (MCK) was 70% efficient (*Mourkioti et al., 2006*), and blocking IKKβ activity using the IKβα super-repressor mouse or transfection of dominant-negative IKKβ completely blocked NFκB nuclear translocation (*Cai et al., 2004*; *Van Gammeren et al., 2009*). The reasons for our limited efficacy are unclear, but may derive from the efficiency of tamoxifen, as ours was the only study to use an inducible transgenic model. However, our lesser impacts on the NFκB pathway are likely more translationally relevant as NFκB inhibitors have a comparable impact on the NFκB pathway as reported here (*Belova et al., 2017*) and have detrimental off-target effects at high doses (reviewed in *Mourkioti and Rosenthal, 2008*). Thus, while it is possible that residual activation of NFκB signaling is sufficient to drive tenotomy-induced atrophy in our model, our data still suggest that pharmacological inhibition of NFκB signaling is not likely to be therapeutically valuable. Furthermore, our findings in wildtype mice that IKKβ protein levels and nuclear NFκB subunits were unchanged by tenotomy provide additional evidence against NFκB driving tenotomy-induced muscle loss as both of these are increased in other models of unloading (*Hunter et al., 2002*; *Van Gammeren et al., 2009*). Furthermore, we also found only a very small increase in the major gene target of NFκB-induced atrophy (MuRF1) following tenotomy, consistent with other reports (*Bialek et al., 2011*; *Liu et al., 2012*).

Evidence for autophagy regulating tenotomy-induced atrophy has been mounting in recent years (*Bialek et al., 2011*; *Gumucio et al., 2012*; *Joshi et al., 2014*; *Ning et al., 2015*; *Hirunsai and Srikuea, 2021*). The evidence presented here supports this contention, but we find surprisingly small effect sizes for all markers investigated. This could be because we did not directly assess autophagic flux and so missed some temporal dynamics since synthesis and degradation are ongoing simultaneously. It could also be because autophagy is only one piece of a more complex atrophic regulation. Notably, Foxo3 regulates autophagy/lysosomal and ubiquitin/proteasomal degradation (*Zhao et al., 2007*), and this master regulator was not assessed in detail here. Additionally, a number of E3 ubiquitin ligases other than MuRF1 and MAFbx regulate muscle atrophy (*Ye et al., 2007*; *Nagpal et al., 2012*; *Hindi et al., 2014*). Thus, it is possible that NFκB-driven atrophy was affected by IKKβ knockdown, but compensation by another atrophic pathway obscured the effects on muscle mass etc. The complex interplay between these pathways warrants further investigation.

Few studies have investigated sex-specific effects in muscle atrophy, particularly in the context of tenotomy. Studies in other disuse models indicated that atrophy, and the pathways that modulate it, may vary by sex. Soleus atrophy during hindlimb unloading was exacerbated in female rats compared with male rats (*Yoshihara et al., 2019*; *Rosa-Caldwell et al., 2021*; *Yoshihara et al., 2022*). This was associated with increased FoxO3a activity, protein ubiquitination, and myostatin expression in females compared to males, with no differences in autophagy or Akt phosphorylation (*Yoshihara et al., 2019*). In humans, females experience greater levels of atrophy in response to disuse and aging, while males have greater loss of muscle in response to cancer (reviewed in *Rosa-Caldwell and Greene, 2019*), suggesting that sex specificity extends to clinical populations as well. In the human RC, symptomatic tear development and progression are more prevalent in males (*Yamamoto et al., 2017*; *Song et al., 2022*). Here, we find that not only do male mice lose more mass after tenotomy, they lose it uniquely through type 2b fiber atrophy, while female mice lose mass uniquely through fiber hypoplasia. It is possible that both sexes experience some fiber degeneration (as has been noted in chronic human RC tears; *Gibbons et al., 2017*), but males are more efficiently repairing the damaged fibers; this speculation would explain the sex specificity in central nuclei, then the load borne by fewer fibers helps maintain fiber CSA. While these differences are relatively minor, they suggest that some sex specificity in chronic tear outcomes could be intrinsically biological or an interaction between intrinsically biological and behavioral factors. To further explore the clinical relevance of these findings, it will be important to determine whether these changes affect recovery of mass and function following surgical repair and rehabilitation. This would necessitate moving to a larger animal model, but would also afford the opportunity to explore whether this sex specificity is conserved between species.

The most dramatic difference we observed between males and females was the increase in basophilic puncta positive for autophagic vesicle markers p62 and LC3 in male muscles at W8 post-tenotomy compared with female muscles. This suggests that the ongoing mass loss concurrent with their appearance may be mediated by autophagy. However, we found only mild differences in LC3 protein content between male and female muscles at this timepoint. Thus, the issue in the male mice may be dysregulated autophagic flux with accumulation of large autophagic vesicles. Proper assessment of autophagic flux cannot be done on existing samples as it requires treatment of mice with lysomotropic agents prior to muscle harvest. Additionally, it is important to note that estrous cycle was not controlled in these mice and sex hormone levels were not measured in this study. These preliminary observations, though intriguing, will require more rigorous follow-up evaluations to define the interaction between sex, tenotomy, and autophagy in naïve wildtype mice.

In conclusion, we found that a twofold gain- or loss-of-function of IKKβ did not impact tenotomy-induced muscle atrophy or contractile dysfunction. This suggests that IKKβ/NFκB inhibitors are unlikely to improve muscle outcomes through direct inhibition of atrophy in human RC tears. Given promising preclinical data suggesting that IKKβ inhibitors improve tendon-to-bone healing after RC tenotomy and repair (*Golman et al., 2021*), and NFκB inhibitors improve muscle mass and contractile function in disease models (*Messina et al., 2006*), it was logical to hypothesize that they would also directly improve muscle outcomes in human rotator cuff disease. However, this hypothesis was not supported by our data, which suggest that tenotomy, and likely human RC tears, is a unique model of muscle atrophy driven by factors associated with the loss of muscle passive tension. It is important to note, however, that NFκB inhibitors still have the potential to improve muscle outcomes indirectly by targeting satellite cells, FAPs, and/or tenocytes, all of which will be involved in the development of pathology or the potential for rehabilitation and none of which were genetically modified in this study. Future work comparing these results with NFκB inhibitors could shed light on their therapeutic potential for preventing tenotomy-induced atrophy and the mechanisms of action. Intriguingly, our data also revealed a preliminary observation of sex specificity in the atrophic response following tenotomy that warrants further exploration. This aligns with sex specificity in human RC tears and could guide more effective and individualized therapies.

# Methods

## Key resources table

| Reagent type (species) or resource | Designation | Source or reference | Identifiers | Additional information |
|---|---|---|---|---|
| Strain, strain background | C57BL/6J: Ikbkbtm2Cgn (mouse) | This paper | Ikbkbtm2Cgn | Originally generated by Manolis Pasparakis (*Pasparakis et al., 2002*) |
| Strain, strain background | C57BL/6J: Ikbkbtm2Cgn (mouse) | Jackson Laboratories | RRID:IMSR_ JAX:008242 | |
| Strain, strain background | C57BL/6J: Tg(ACTA1-cre/ Esr1*)2Kesr/J (mouse) | Jackson Laboratories | RRID:IMSR_ JAX:025750 | |
| Antibody | Anti-myosin heavy chain (slow, alpha-, and beta-) (mouse monoclonal) | Developmental Studies Hybridoma Bank | BA-F8 | IF (1:30) |
| Antibody | Anti-myosin heavy chain type IIA (mouse monoclonal) | Developmental Studies Hybridoma Bank | SC-71 | IF (1:30) |
| Antibody | Anti-myosin heavy chain type IIB (mouse monoclonal) | Developmental Studies Hybridoma Bank | BF-F3 | IF (1:30) |
| Antibody | anti-laminin (rabbit polyclonal) | Abcam | 11575 | IF (1:400) |
| Antibody | Anti- SQSTM1/p62 (rabbit polyclonal) | Cell Signaling Technology | 5114 | IF (1:100), WB (1:1000) |
| Antibody | Anti-LC3A/B (rabbit monoclonal) | Cell Signaling Technology | 12741 | IF (1:100), WB (1:1000) |

*Continued on next page*

*Continued*

| Reagent type (species) or resource | Designation | Source or reference | Identifiers | Additional information |
|---|---|---|---|---|
| Antibody | Anti-PDGFRα (goat polyclonal) | R&D Systems | AF1062 | IF (1:400) |
| Antibody | Anti-Pax7 (mouse monoclonal) | Developmental Studies Hybridoma Bank | Pax7 | IF (1:100) |
| Antibody | Anti-IKKβ (rabbit monoclonal) | Cell Signaling Technology | 8943 | WB (1:1000) |
| Antibody | Anti-mTOR (rabbit monoclonal) | Cell Signaling Technology | 2983 | WB (1:1000) |
| Antibody | Anti-phospho-mTOR (Ser2448) (rabbit monoclonal) | Cell Signaling Technology | 5536 | WB (1:1000) |
| Antibody | Anti-Akt (rabbit polyclonal) | Cell Signaling Technology | 9272 | WB (1:1000) |
| Antibody | Anti-phospho-Akt (Ser473) (rabbit monoclonal) | Cell Signaling Technology | 4060 | WB (1:1000) |
| Antibody | Anti-ubiquitin (rabbit polyclonal) | Dako | Z0458 | WB (1:1000) |
| Antibody | Anti-actin (rabbit polyclonal) | Sigma-Aldrich | A2066 | WB (1:10,000) |
| Antibody | Anti-NF-κB p65 (rabbit monoclonal) | Cell Signaling Technology | 8242 | WB (1:1000) |
| Antibody | Anti-NF-κB1 p105/p50 (rabbit monoclonal) | Cell Signaling Technology | 13586 | WB (1:1000) |
| Antibody | Anti-histone H3 (rabbit monoclonal) | Cell Signaling Technology | 4499 | WB (1:1000) |
| Antibody | Anti-GAPDH (rabbit polyclonal) | Abcam | 9485 | WB (1:10,000) |
| Commercial assay or kit | Nuclear Extract Kit | Active Motif | 40010 | |
| Commercial assay or kit | M.O.M. Mouse-on-moue Kit | Vector Laboratories | BMK-2202 | |
| Commercial assay or kit | MultiScribe reverse transcription kit | Applied Biosystems | 4368814 | |
| Chemical compound, drug | Antigen Retrieval Citra (pH-6.0) | Biogenex | HK086-5K | |
| Software, algorithm | ImageJ | ImageJ | | |
| Software, algorithm | GraphPad Prism | GraphPad Prism | | |
| Software, algorithm | Matlab | MathWorks | | |
| Software, algorithm | Image Studio | LI-COR | | |
| Sequence-based reagent | Fbxo32_F | IDT | qPCR primers | *AACCGGGAGGCCAGCTAAAGAACA* |
| Sequence-based reagent | Fbxo32_R | IDT | qPCR primers | *TGGGCCTACAGAACAGACAGTGC* |
| Sequence-based reagent | Trim63_F | IDT | qPCR primers | *GAGAACCTGGAGAAGCAGCT* |
| Sequence-based reagent | Trim63_R | IDT | qPCR primers | *CCGCGGTTGGTCCAGTAG* |
| Sequence-based reagent | Mstn_F | IDT | qPCR primers | *CAGACCCGTCAAGACTCCTACA* |

*Continued on next page*

*Continued*

| Reagent type (species) or resource | Designation | Source or reference | Identifiers | Additional information |
|---|---|---|---|---|
| Sequence-based reagent | Mstn_R | IDT | qPCR primers | *CAGTGCCTGGGCTCATGTCAAG* |
| Sequence-based reagent | Foxo3_F | IDT | qPCR primers | *ATCGCCTCCTGGCGGGCTTA* |
| Sequence-based reagent | Foxo3_R | IDT | qPCR primers | ACGGCGGTGCTAGCCTGAGA |
| Sequence-based reagent | Lc3b_F | IDT | qPCR primers | *cactgctctgtcttgtgtaggttg* |
| Sequence-based reagent | Lc3b_R | IDT | qPCR primers | *tcgttgtgcctttattagtgcatc* |
| Sequence-based reagent | Gabarapl1_F | IDT | qPCR primers | *catcgtggagaaggctccta* |
| Sequence-based reagent | Gabarapl1_R | IDT | qPCR primers | *atacagctggcccatggtag* |
| Sequence-based reagent | Bnip3_F | IDT | qPCR primers | *AGGGCTCCTGGGTAGAACTG* |
| Sequence-based reagent | Bnip3_R | IDT | qPCR primers | *GCTGGGCATCCAACAGTATT* |
| Sequence-based reagent | Becn1_F | IDT | qPCR primers | *AGCCTCTGAAACTGGACACG* |
| Sequence-based reagent | Becn1_R | IDT | qPCR primers | *CCTCTTCCTCCTGGGTCTCT* |
| Sequence-based reagent | Atg5_F | IDT | qPCR primers | *ggagagaagaggagccaggt* |
| Sequence-based reagent | Atg5_R | IDT | qPCR primers | *gctgggggacaatgctaata* |
| Sequence-based reagent | Gapdh_F | IDT | qPCR primers | *TGTGATGGGTGTGAACCACGAGAA* |
| Sequence-based reagent | Gapdh_R | IDT | qPCR primers | *GAGCCCTTCCACAATGCCAAAGT* |

## Study approval

All animal work described was performed in accordance with the National Institutes of Health's Guide for the Use and Care of Laboratory Animals and was approved by the Animal Studies Committee of the Washington University School of Medicine (IACUC 20-0459).

## Experimental design

Experiments were performed on male and female muscle-specific conditional IKKβ knockdown mice (IKKb[MKD]), muscle-specific conditionally active IKKβ mice (IKKb[MCA]), and respective littermate controls at 6–8 months of age. IKKβ-MKD mice were generated by breeding Ikbkbtm2Cgn (originally generated by Manolis Pasparakis; *Pasparakis et al., 2002*) mice with HSA-MCM (Jackson Labs; #025750) mice such that tamoxifen delivery induced deletion of the IKKβ gene in mature muscle fibers. IKKβ-CA mice were generated by breeding Gt(Rosa26)tm4(Ikbkb)Rsky (Jackson Labs; #008242) mice with HSA-MCM mice such that tamoxifen delivery induced expression of a constitutively active form of IKKβ in mature muscle fibers. Littermate mice that did not express Cre-recombinase were used as wildtype (WT) controls. Some data from WT mice have been reported previously (*Meyer, 2022*). Seven mice per group were randomly assigned in each design – IKKb[MKD] (three timepoints, two treatments, two genotypes, two sexes) and IKKb[MCA] (one timepoint, two treatments, two genotypes, two sexes) – for a total of 224 mice. Numbers per group for each outcome measure are provided for each figure and deviate from 7 in cases where muscles failed during testing, sample quality was poor or insufficient sample remained for the assay. Each mouse underwent tamoxifen treatment, bilateral tenotomy, or

sham surgery followed by ex vivo muscle contractile testing and sacrifice. Muscles were then processed for additional measurements as described below: the SS from the right shoulder and the IS from the left shoulder were used for physiological testing and the SS from the left shoulder and the IS from the right shoulder were prepared for histology, qPCR, and Western blotting.

## Surgical treatment

Under continuous anesthesia (2% inhaled isoflurane at 2 L/min), both shoulders were prepared for sterile surgery. In a left lateral decubitus position, an ~3 mm incision was made through the skin from the acromial arch toward the mid-belly of the deltoid muscle, exposing the proximal deltoid. An incision was then made through the deltoid to expose the humerus. The humerus was then stabilized with one set of forceps while the acromial space was expanded with another forceps. Mice in the tenotomy group then had the tendons of the SS and IS muscles transected with microscissors. Mice in the sham group had no tendon transection. All mice then had the deltoid sutured to the trapezius muscle over the acromial arch with 5-0 absorbable Vicryl suture (Ethicon) and the skin incision closed with Vetbond suture glue (3M). The mice were then turned to lie on the right side and the same procedure was repeated for a bilateral treatment. This design was chosen to avoid the effects of altered use that could confound a contralateral control in a unilateral design. Following surgery, mice were provided analgesia and allowed free cage activity until the experimental end point.

## Ex vivo muscle contractile testing

Mice assigned to the 2- and 8-week post-tenotomy outcome groups were anesthetized again at the experimental end point for ex vivo contractile testing of the SS and IS muscles as previously described (JOR and JCSM). The SS muscle was excised from the right side with its connection to the scapula and humeral head left intact. All other musculature was dissected free from the scapula, including the IS, which was prepared for histology as described below. The SS was then transferred to an ex vivo physiology rig (Aurora Scientific; 1300A) where it was immersed in Mammalian Ringers solution with the scapula secured to the arm of a dual force/length transducer (Aurora Scientific; 305C-LR) and the humeral head secured to a rigid post. Contraction was elicited by stimulation through parallel plate electrodes flanking the muscle. Muscle length was increased incrementally until the force of twitch contractions began to plateau. Then optimal length was determined by further adjusting length until peak tetanic tension was achieved. Muscle optimal length was then measured with a flexible ruler and the muscle was dissected from bony attachments, blotted, weighed, and flash-frozen for qPCR. All forces were normalized to physiological CSA, which was calculated from model-predicted fiber length and pennation angle values (**Meyer, 2022**). A similar procedure was then repeated on the left side for contractile testing of the IS and histological prep of the SS. Following testing, mice were euthanized. Mice assigned to the 1-week post-tenotomy outcome group were euthanized without contractile testing as the connective tissue 'pseudo-tendon' was too weak to sustain the connection to the humeral head during testing. Muscles that would have undergone contractile testing were flash-frozen in liquid nitrogen and the other muscles were prepared for histology as they were for the other groups.

## Histological measurements

Muscles prepared for histology were affixed to cork at the distal end with tragacanth gum and frozen in liquid nitrogen cooled isopentane and stored at –80°C until sectioning. They were then transferred to a cryostat (Leica; CM1950) where they were transected at mid-belly with a razor blade. The proximal tip was stored at –80°C for Western blot and qPCR assays. The remainder was then sectioned at the mid-belly face at 10 µm for H&E, picrosirius red (Sirius Red), and Oil Red-O (ORO) staining and immunostaining for fiber typing by myosin heavy chain isoform (Developmental Studies Hybridoma Bank; BA-F8, SC-71, BF-F3; 1:30), fiber counting by laminin (Abcam; 11575; 1:400) outline, autophagic vesicle identification by p62 (Cell Signaling Technology [CST]; 5114; 1:100) and LC3A/B (CST; 12741; 1:100), FAP identification by platelet-derived growth factor α (PDGFRα) (R&D Systems; AF1062, 1:400), and satellite cells (SC) by Pax7 (Developmental Studies Hybridoma Bank; Pax7, 1:100).

Sections for ORO staining were mounted on pre-chilled slides and vapor-fixed for 48 hr in formaldehyde at –20°C. They were then post-fixed by immersion in 37% formaldehyde for 30 min. For staining, slides were transferred to a working solution of ORO consisting of a 3:2 ratio of ORO stock

(5 mg/mL in isopropanol) and diH$_2$O for 10 min with agitation. They were then rehydrated and rinsed in diH$_2$O for imaging. Quantification of fatty infiltration was performed on 2× images of the entire cross-section. Sections for Sirius Red staining were fixed in acetone for 1 hr, followed by Bouin's solution for 5 min. Slides were rinsed in running distilled water for 10 min and then immersed in a freshly made Sirius Red solution (1 mg/mL Direct Red 80 dissolved in saturated picric acid at 37°C, cooled to room temperature) for 2 hr. Slides were then placed in 0.01 M hydrochloric acid for 5 min, rinsed in distilled water for 1 min, dehydrated in ethanol, cleared in xylenes, and coverslipped. Quantification of fibrosis was performed on two 10× images that excluded the tendon and averaged. Images of ORO and Sirius Red were thresholded on the red channel by the Huang algorithm in ImageJ (NIH). Fatty infiltration and fibrosis were then quantified as the percentage of total pixels in the cross-section that met the threshold.

Immunostaining for p62, LC3A/B, PDGFRα, and Pax7 was performed on sections fixed in 4% paraformaldehyde for 15 min. Slides allocated for Pax7 underwent an additional step of antigen retrieval consisting of steaming in pre-heated Antigen Retrieval Citra (Biogenex; HK086-5K) for 15 min, followed by gradual cooling to room temperature and blocking with the M.O.M. (Mouse on Mouse) kit (Vector Laboratories) according to the manufacturer's instructions. Slides allocated for p62 and LC3A/B were stained with H&E following imaging, allowing positive identification of p62/LC3 and hematoxylin in the same section. All other immunostaining was performed on fresh-frozen sections. With the exception of myosin heavy chain-stained sections, all immunostaining was counterstained with Hoechst 33342 (DAPI). Fiber area by fiber type was quantified on four 20× images from defined regions of the superficial and deep portion of the muscle using a semi-automated ImageJ macro. First, fiber ROIs are identified by thresholding the laminin signal using the Huang algorithm followed by selecting outlined regions by the Analyze Particles algorithm with manual deletion/addition of incorrect/missing regions. Then ROIs are typed by overlaying ROIs on channels stained with each myosin heavy chain type sequentially and scored as positive or negative by average signal. Type assignment is checked and errors corrected manually. Fiber number was quantified on 2× laminin stained images of the entire cross-section using the ROI identification portion of the algorithm. Central nuclei were defined as structures positive for hematoxylin and DAPI not adjacent to a laminin boundary and were counted manually through the entire cross-section. Basophilic puncta were defined as structures positive for hematoxylin but negative for DAPI and were counted manually through the entire cross-section. FAPs and SCs were counted manually on two nonoverlapping 10× images. FAPs were defined as structures positive for PDGFRα in the interstitium between laminin fiber boundaries. SCs were defined as Pax7-positive nuclei beneath the laminin boundary.

## Quantitative real-time PCR

RNA was extracted from frozen muscle using a standard Trizol/chloroform extraction protocol. Briefly, muscle was bead homogenized in Trizol using a TissueLyser II (QIAGEN; 85300) and RNA extracted by addition of chloroform with centrifugation. RNA was precipitated in 50% isopropanol, washed in 75% ethanol, and dissolved in DNAse/RNAse free water. cDNA was generated using the MultiScribe reverse transcription kit (Applied Biosystems; 4368814) according to the manufacturer's instructions. Transcript copies were detected using a fast SYBR Green PCR master mix (Applied Biosystems; 4385612). Primer sequences are listed in the Key Resources Table. Reactions were run in duplicate on a QuantStudio3 (Applied Biosystems) real-time PCR system. All expression values were normalized to GAPDH, and primers were validated against a positive control (*Figure 2—figure supplement 1*, *Figure 7—figure supplements 1 and 2*).

## Western blotting

Protein was extracted from frozen muscle in RIPA buffer supplemented with cOmplete Protease Inhibitor (Roche) with bead homogenization using a TissueLyser II (QIAGEN; 85300). Homogenized tissue was then solubilized for 1 hr at 4°C with agitation, centrifuged, and the protein concentration of the supernatant determined by a Pierce BCA assay (Thermo Fisher Scientific; 23225) according to the manufacturer's instructions. Then, equivalent amounts of protein (40 µg) diluted in diH$_2$O with Laemmli buffer were denatured and separated on 4–12% Bis-Tris gels (Invitrogen; NW04120). Protein was then transferred to polyvinylidene difluoride (PVDF) membrane, reversibly stained with Ponceau S and blocked in TBST+ (1× Tris-buffered saline with 2.5% fish gelatin, 0.1% sodium azide, and 0.5% tween).

The following primary antibodies were applied overnight at 4°C at 1:1000 unless otherwise noted: IKKβ (CST; 8943), ubiquitin (Dako; Z0458), SQSTM1/p62 (CST; 5114), LC3A/B (CST; 12741), mTOR (CST; 2983), phospho-mTOR-ser2448 (CST; 5536), Akt (CST; 9272), phospho-Akt-ser473 (CST; 4060), and actin (Sigma; A2066; 1:10,000). To assay NFκB subunit translocation, protein from a subset of muscles was fractionated into nuclear and cytoplasmic components using a Nuclear Extract Kit (Active Motif; 40010) according to the manufacturer's instructions. The nuclear fraction was validated against an established protocol (*Dimauro et al., 2012*) and separated, transferred, and blocked as described above. The following primary antibodies were applied overnight at 4°C at 1:1000 unless otherwise noted: p65 (CST; 8242), p50/105 (CST; 13586), Histone H3 (CST; 4499), and GAPDH (Abcam; 9485; 1:10,000). Following incubation, membranes were washed and incubated for 1 hr with the relevant secondary antibodies and imaged with a LI-COR Odyssey. Blot analysis was performed using Image Studio (LI-COR). Band intensities were normalized to an actin loading control with the exception of p50 and p65, which were normalized to histone H3 and ubiquitin that was normalized to Ponceau. When possible, antibodies were validated with a positive loading control (*Figure 2—figure supplement 1*, *Figure 7—figure supplements 1 and 2*).

## Prediction of mass deficit from morphological measurements

The relative contribution of fiber CSA, length, and number changes to the mass deficit was calculated by modeling fibers as cylinders. Under this assumption, the mass loss in a fiber ($\Delta m$) resulting from a decrease in fiber length ($\Delta FL$) and decrease in fiber CSA ($\Delta F_{CSA}$) was calculated by multiplying the two and the density of muscle, $\rho$ (*Ward and Lieber, 2005*). Then, the mass loss in the muscle was calculated by multiplying the mass loss in individual fiber types by the relative fraction of each fiber type (*Rui et al., 2016*) and the decrease in fiber number ($\Delta FN$), according to *Equation 1*. This equation assumes that the density of muscle and the relative fraction of each fiber type is unchanged following tenotomy:

$$\Delta m = \Delta FN \cdot \left( \left( \Delta CSA_1 \cdot AF_1 \right) + \left( \Delta CSA_{2a} \cdot AF_{2a} \right) + \left( \Delta CSA_{2x} \cdot AF_{2x} \right) + \left( \Delta CSA_{2b} \cdot AF_{2b} \right) \right) \cdot \Delta FL \cdot \rho \quad (1)$$

## Statistical analyses and reproducibility

Sample size was selected a priori using a power calculation (G*Power) with variance estimated from measures of SS mass and peak tetanic tension from a previous study. With n = 7, this study was predicted to be able to detect a 15% difference in sample means of both variables with $\alpha = 0.05$ and $(1-\beta) = 0.8$. Following genotyping and nonrandomized group assignment, mice were given a numeric identifier and all experimenters were blinded to genotype. Cage-mates were assigned to a single treatment to enable true littermate controls. Between-group comparisons were made by two- or three-way ANOVA as indicated, with Sidak's or FDR multiple testing correction applied, respectively. Normality was tested using the Shapiro–Wilk test. Actual numbers per group are indicated in each figure for each analysis. All results are presented as mean ± standard deviation. All statistical analyses were performed with GraphPad Prism.

## Acknowledgements

Special thanks to Kathryn Bohnert and Heangun Yoon for technical assistance and Dr. Andrew Findlay for insightful advice and discussion.

## Additional information

### Competing interests

Stavros Thomopoulos: Reviewing editor, *eLife*. The other authors declare that no competing interests exist.

## Funding

| Funder | Grant reference number | Author |
|---|---|---|
| National Institute of Arthritis and Musculoskeletal and Skin Diseases | R21AR071582 | Gretchen A Meyer |
| National Institute of Arthritis and Musculoskeletal and Skin Diseases | R01AR057836 | Gretchen A Meyer Stavros Thomopoulos |

The funders had no role in study design, data collection and interpretation, or the decision to submit the work for publication.

## Author contributions

Gretchen A Meyer, Conceptualization, Resources, Data curation, Software, Formal analysis, Supervision, Funding acquisition, Validation, Investigation, Visualization, Methodology, Writing – original draft, Project administration, Writing – review and editing; Stavros Thomopoulos, Conceptualization, Funding acquisition, Project administration, Writing – review and editing; Yousef Abu-Amer, Resources, Validation, Methodology, Writing – review and editing; Karen C Shen, Data curation, Writing – review and editing

## Author ORCIDs

Gretchen A Meyer ⓘ http://orcid.org/0000-0001-9268-3993
Yousef Abu-Amer ⓘ http://orcid.org/0000-0002-5890-5086

## Ethics

All animal work described was performed in accordance with the National Institutes of Health's Guide for the Use and Care of Laboratory Animals and was approved by the Animal Studies Committee of the Washington University School of Medicine (IACUC 20-0459).

## Decision letter and Author response

Decision letter https://doi.org/10.7554/eLife.82016.sa1
Author response https://doi.org/10.7554/eLife.82016.sa2

---

## Additional files

### Supplementary files

• Transparent reporting form

### Data availability

All data generated or analyzed during this study are included. Data files have been provided for all Figures.

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
