## [Editor Report]

This articlce challenged the premise that NF-kappaB and its upstream kinase IKKbeta play a role in muscle atrophy following tenotomy. Two animal models were used – one leading to enhanced muscle-specific NF-kappaB activation and the other a muscle-specific deletion. In both models, there was no significant relationship to observed muscle changes following tenotomy. Overall, this work is significant in that it challenges the existing dogma that NF-kappaB plays a crucial role in muscle atrophy.

---

## [Decision Letter]

**Decision letter after peer review:**

Thank you for submitting your article "Tenotomy-induced muscle atrophy is sex-specific and independent of NFκB" for consideration by *eLife*. Your article has been reviewed by 3 peer reviewers, one of whom is a member of our Board of Reviewing Editors, and the evaluation has been overseen by a Reviewing Editor and Mone Zaidi as the Senior Editor. The following individual involved in the review of your submission has agreed to reveal their identity: Brian Feeley (Reviewer #2).

Essential revisions:

Overall the balance of the reviews is positive and suggests that there are novel findings worthy of publication.

1) Please carefully review the very helpful and constructive comments by the three reviewers and address them in detail.

2) Reviewer three notes the role of satellite cells and stromal progenitors (FAPs) in injury response to rotator cuff tenotomy is lacking; please address this to the reviewer's satisfaction.

*Reviewer #1 (Recommendations for the authors):*

Overall this is an outstanding manuscript with a few revisions recommended.

1) A comparison of the effects of the knockout relative to a wild-type control treated with agents to inhibit the IKK/NFkB pathway would have been very helpful to advance our understanding of potential therapeutic intervention as well as validation of the mouse models.

2) For figure 4, it would be helpful to understand why there are no observed changes in contractility and tension if there are significant physical differences in muscle composition observed.

3) In Figure 6G the double arrows are very difficult to see. Similar to E, it would be better to increase their size.

*Reviewer #2 (Recommendations for the authors):*

Overall, excellent job with 'negative data' and a very comprehensive, thorough approach to proving this information. I found the results partially gratifying since we found similar findings many years ago (albeit with much less refined mechanisms!).

The science is well presented in the introduction, with a strong rationale shown for the choice of NF-κB in regulating atrophy in this model. In LIne 77, it would be great to support the Bialek study earlier with other differences in atrophy, but not absolutely required. Again, although not necessary, a figure outlining the NF-κB pathway in muscle atrophy would be nice given its overall complexity.

The hypothesis is clear and well stated, and the primary conclusions were stated well, but I would change 'it' in Line 105 to 'muscle atrophy' to avoid any confusion if the reader is going quickly.

The results are clear, but my bias is that the 'time x sex ANOVA' sections take away from the reading and make it more complicated to follow what are a clear set of experiments. The graphs are clearly presented and make it clear that any differences between groups are subtle, which I think is clearly presented in the paper as well.

Line 193--I would say 'no statistical difference' given that the P did not approach 0.05 and the overall #'s were small. I prefer that over 'trend' and then just report the P value and let the reader interpret it.

The Discussion is well done and addresses the limitations of the model as well as its strengths of the model. In particular, LInes 372-380 highlight the influence of IKKB vs NF-κB which shows the complicated nature of this pathway.

I would like to see different limitations--the timepoints are standard for this model and represent chronic conditions. I'd explore in more depth LInes 455-456--the IKKB vs NF-κB differences and compensation-based pathways, and address other potential hypotheses in a bit more detail.

*Reviewer #3 (Recommendations for the authors):*

1. Authors should cite original research articles, not review papers that summarize disuse muscle atrophy. For example, the original research on MAFbx and *Murf1* should be cited.

2. Authors are repeatedly using the word 'architectural' but 'morphological' is a more suitable word to describe muscle plasticity or adaptation.

3. In rotator cuff injury, whether you have denervation in SS and/or IS muscle makes a significant difference in severity and recovery. Authors should address this.

4. In the mature skeletal muscle, hyperplasia, and hypoplasia do not occur frequently. Data supporting these observations are lacking. For this to happen, strong activation of muscle stem cells is required.

5. Authors should measure or at least discuss the role of satellite cells and stromal progenitors (FAPs) in injury response to rotator cuff tenotomy. The regenerative response and NFkB signaling will most likely affect muscle stem cells and niche cells more than mature fibers, especially in the week 2 time period. Accretion of satellite cell-derived myonuclei during regeneration will not have genetic modifications of NF-κB. A single muscle stem cell could give rise to hundreds of myonuclei, potentially diluting the effects. This is a major limitation of the current study.

6. It has been previously documented that NF-κB is not the only catabolic signal that activates ubiquitin-proteasome and lysosomal autophagy. Foxo3 can regulate both UPS and autophagy. Moreover, there are several E3 ligases other than MAFbx1 and *Murf1* that play a role in muscle atrophy conditions.

---

## [Author Response]

Reviewer #1 (Recommendations for the authors):Overall this is an outstanding manuscript with a few revisions recommended.1) A comparison of the effects of the knockout relative to a wild-type control treated with agents to inhibit the IKK/NFkB pathway would have been very helpful to advance our understanding of potential therapeutic intervention as well as validation of the mouse models.

Thanks for this suggestion and as noted above, we agree that this would enhance our understanding of the general therapeutic benefit of NFkb inhibitors for the treatment of RC injuries. However, we believe that pharmacologics are likely to impact other aspects of RC tenotomy pathology (e.g. tendon healing, which has already been documented) that will indirectly affect muscle atrophy. Because of the number of potential indirect mediators and their interconnectedness, this quickly becomes a complicated question. We believe this question would be better suited for an additional study, where we can apply what we have learned here using genetic tools about the role of NFkb in the muscle fiber specifically.

We feel this is an important consideration for the reader and thus have added the following to the Discussion section:

Discussion:

“It is important to note, however, that NFκB inhibitors still have the potential to improve muscle outcomes indirectly by targeting satellite cells, fibro/adipogenic progenitors and/or tenocytes, all of which will be involved in the development of pathology or the potential for rehabilitation and none of which were genetically modified in this study. Future work comparing these results with NFκB inhibitors could shed light on their therapeutic potential for preventing tenotomy-induced atrophy and the mechanisms of action.”

2) For figure 4, it would be helpful to understand why there are no observed changes in contractility and tension if there are significant physical differences in muscle composition observed.

We are happy to provide a more detailed discussion of the contractility data and the potential physical explanators. Theoretically, normalization of tetanic force to physiological cross-sectional area (PCSA) in Fig 4 accounts for our measured changes in mass (Fig 3) and fiber length (Fig 5F), and by proxy fiber atrophy and hypoplasia (Fig 5D&E). However, as you noted, there is a normalized tension deficit in tenotomized muscle (compared with sham) and it does not advance between weeks 2 and 8 despite notable changes in fiber structure during this timeframe (Fig 6E-H). We speculate that the loss of normalized tension happens early in tenotomy (it has been documented as early as day one) and is not fully recovered by the subsequent processes of healing. It is difficult to say whether some recovery of this initial mechanism occurs and is offset by the later development of structural pathology (e.g. fibrosis, fat infiltration, basophilic puncta etc.) or whether the degree of this later structural pathology is insufficient to drive additional measurable contractile changes. Future studies using targeted mouse models such as the lipodystrophic mouse (that does not develop fat infiltration) or pharmacologics that target accumulation of autophagic vesicles could shed some light on this question.

To clarify these connections for the reader we have added the following to the Results and Discussion sections:

Results:

“This suggests that normalizing contractile tension to PCSA fully accounts for the ongoing mass loss.”

Discussion:

“This suggests that the contractile deficit originated early in the tenotomy injury and never resolved during the subsequent process of healing. There is considerable evidence for myofibrillar disruption and fiber damage post tenotomy (reviewed in (Jamali *et al.*, 2000)), and it is possible that the tenotomized muscle cannot adequately repair this damage in its unloaded state. If this is the case, it also offers an explanation for why IKKβ knockdown did not affect the contractile deficits. However, it is difficult to say whether some recovery of this initial damage occurs and is offset by the later development of structural pathology (e.g. fibrosis, fat infiltration, basophilic puncta etc.) or whether the degree of this later structural pathology is insufficient to drive additional measurable contractile changes. Future studies using targeted mouse models such as the lipodystrophic mouse (that does not develop fat infiltration) or pharmacologics that target accumulation of autophagic vesicles could shed some light on this question.”

3) In Figure 6G the double arrows are very difficult to see. Similar to E, it would be better to increase their size.

Thank you for noting this. We agree and the arrows have been enlarged.

Reviewer #2 (Recommendations for the authors):Overall, excellent job with 'negative data' and a very comprehensive, thorough approach to proving this information. I found the results partially gratifying since we found similar findings many years ago (albeit with much less refined mechanisms!).The science is well presented in the introduction, with a strong rationale shown for the choice of NF-κB in regulating atrophy in this model. In LIne 77, it would be great to support the Bialek study earlier with other differences in atrophy, but not absolutely required. Again, although not necessary, a figure outlining the NF-κB pathway in muscle atrophy would be nice given its overall complexity.

Thanks for these suggestions. We were unable to find another comparative atrophic signaling study that included tenotomy to support the Bialek study, but are happy to include one if you could point us to it. While we agree that the NFkb pathway is indeed complex, we would prefer to direct the reader to an excellent review (Jackman et al., 2013) with figures detailing the pathway. We have now explicitly noted that a pathway diagram can be found there.

The hypothesis is clear and well stated, and the primary conclusions were stated well, but I would change 'it' in Line 105 to 'muscle atrophy' to avoid any confusion if the reader is going quickly.

Thank you for this suggestion. “It” has now been changed to “muscle atrophy.”

The results are clear, but my bias is that the 'time x sex ANOVA' sections take away from the reading and make it more complicated to follow what are a clear set of experiments. The graphs are clearly presented and make it clear that any differences between groups are subtle, which I think is clearly presented in the paper as well.

We agree that listing ANOVA results can bog down the reader and detract from interpretation. We have removed or edited superfluous ANOVA results from the text except for the occasions where we feel they enhance the presentation of the results.

Line 193--I would say 'no statistical difference' given that the P did not approach 0.05 and the overall #'s were small. I prefer that over 'trend' and then just report the P value and let the reader interpret it.

This change has been made to line 193 to eliminate reference to the “trend.”

The Discussion is well done and addresses the limitations of the model as well as its strengths of the model. In particular, LInes 372-380 highlight the influence of IKKB vs NF-κB which shows the complicated nature of this pathway.

Thank you.

I would like to see different limitations--the timepoints are standard for this model and represent chronic conditions. I'd explore in more depth LInes 455-456--the IKKB vs NF-κB differences and compensation-based pathways, and address other potential hypotheses in a bit more detail.

Thanks for this suggestion. We have expanded the Discussion in this area to include more detail on other pathways that could regulate atrophy independent of or in interaction with NFkb.

Discussion:

“It could also be because autophagy is only one piece of a more complex atrophic regulation. Notably, Foxo3 regulates autophagy/lysosomal and ubiquitin/proteasomal degradation (Mammucari *et al.*, 2007) and this master regulator was not assessed in detail here. Additionally, a number of E3 ubiquitin ligases other than *Murf1* and MAFbx regulate muscle atrophy (Ye *et al.*, 2007; Nagpal *et al.*, 2012; Hindi *et al.*, 2014). Thus, it is possible that NFκB-driven atrophy was affected by IKKβ knockdown, but compensation by another atrophic pathway obscured the effects on muscle mass etc. The complex interplay between these pathways warrants further investigation.”

Reviewer #3 (Recommendations for the authors):1. Authors should cite original research articles, not review papers that summarize disuse muscle atrophy. For example, the original research on MAFbx and Murf1 should be cited.

We appreciate this suggestion. We have reviewed the manuscript and deleted extraneous citations of reviews where an original article reference could be substituted to support the statement. Original articles are cited alongside the review papers when appropriate, in particular the original research on the discovery of MAFbx and *Murf1* are highlighted (Bodine et al., 2001; Gomes et al., 2001). We decided to retain review references where we believe a review is the best citation (e.x. *Jackman et al.* for a more detailed summary with the pathway diagram requested by Reviewer 2, *Paris-Moreno et al.* and *Laron et al.* that make the statement our sentence was alluding to).

2. Authors are repeatedly using the word 'architectural' but 'morphological' is a more suitable word to describe muscle plasticity or adaptation.

Architectural has been changed to morphological throughout the manuscript.

3. In rotator cuff injury, whether you have denervation in SS and/or IS muscle makes a significant difference in severity and recovery. Authors should address this.

We completely agree and this is an excellent suggestion. We have clarified in the introduction and discussion that our model involves no denervation. We also note that this (1) isolates tenotomy-induced atrophy from denervation-induced atrophy, which is important because denervation-induced atrophy is mediated by NFkb and (2) is arguably more representative of human rotator cuff injury which does not consistently involve nerve damage.

Introduction:

“…tenotomy of the rotator cuff (RC) muscles (without suprascapular nerve injury)”

Discussion:

“Importantly, tenotomy alone does not involve any direct damage to the muscle or its innervation, avoiding confounding effects of denervation and extensive regeneration. While RC tenotomy is frequently paired with denervation to elicit more dramatic muscle pathology (particularly fatty infiltration), there is not strong evidence that nerve damage is a consistent feature of even massive RC tears in humans (Costouros *et al.*, 2007).”

4. In the mature skeletal muscle, hyperplasia, and hypoplasia do not occur frequently. Data supporting these observations are lacking. For this to happen, strong activation of muscle stem cells is required.

We agree for hyperplasia. There is not compelling evidence that addition of myofibers happens even in the context of extensive hypertrophy and would absolutely require satellite cell participation. Along these lines, the evidence we present in this work does not support hyperplasia in response to tenotomy. While fibers with central nuclei are mildly increased at later stages of tenotomy-induced atrophy, we believe these to be the result of repair or regeneration events rather than hyperplasia.

However, our evidence does support hypoplasia. While myofiber count is a somewhat crude measurement, is has been used extensively to document hypoplasia in aging rodent and human muscle and associated with some myopathies. Though it hasn’t been directly demonstrated following tenotomy, there is strong evidence for myofiber degeneration and apoptosis in muscle from chronically torn human rotator cuffs (PMID 28145949).

To clarify our hypothesis that muscles are engaging in fiber repair (or locally regeneration) rather than hyperplasia, we have modified our discussion of centralized nuclei as follows:

Discussion:

“It is possible that both sexes experience some fiber degeneration (as has been noted in chronic human RC tears (Gibbons et al., 2017)), but males are more efficiently repairing the damaged fibers; this speculation would explain the sex-specificity in central nuclei, then the load borne by fewer fibers helps maintain fiber CSA.”

“Similarly, extensive regeneration does not appear to be a notable feature of RC tears in humans, where only ~10% of fibers exhibit central nucleation, compared with ~3% in controls (Gibbons *et al.*, 2017). In the mouse, we find no proliferation of satellite cells at W1 following tenotomy and only a minor (<3%) increase in centrally nucleated fibers at later timepoints. This, combined with the lack of dilution of the NFkb knockdown at the gene and protein level, suggests that the genetically unmodified SCs are not contributing substantial additional myonuclei.”

5. Authors should measure or at least discuss the role of satellite cells and stromal progenitors (FAPs) in injury response to rotator cuff tenotomy. The regenerative response and NFkB signaling will most likely affect muscle stem cells and niche cells more than mature fibers, especially in the week 2 time period. Accretion of satellite cell-derived myonuclei during regeneration will not have genetic modifications of NF-κB. A single muscle stem cell could give rise to hundreds of myonuclei, potentially diluting the effects. This is a major limitation of the current study.

Thank you for this suggestion. Your point is well taken that satellite cells and FAPs are likely active in the early phases of tenotomy-induced atrophy and will not have IKKb knockdown. To determine whether the numbers of satellite cells and FAPs are altered by tenotomy and/or IKKb knockdown, we have quantified these cell populations via histology at week 1 and added this data to Figure 6.

Based on these data and consistent with our other data, we believe that the regenerative response following tenotomy is fairly limited. The direct injury of tenotomy is to the tendon and secondary injury to the myofibers upon retraction doesn’t appear to engage satellite cells in extensive regeneration. We base this conclusion primarily on our histological findings at week 1 which show no change in Px7+ myonuclei per fiber and less than 1% fibers as centrally nucleated (Figure 6F). Additional support for accretion of satellite cell derived nuclei being limited comes from our measures of IKKb gene and protein expression which are equivalent between sham and tenotomy (Figure 2A). To clarify this point for the reader, we have added the following text to the Discussion:

Discussion:

“Similarly, extensive regeneration does not appear to be a notable feature of RC tears in humans, where only ~10% of fibers exhibit central nucleation, compared with ~3% in controls (Gibbons *et al.*, 2017). In the mouse, we find no proliferation of satellite cells at W1 following tenotomy and only a minor (<3%) increase in centrally nucleated fibers at later timepoints. This, combined with the lack of dilution of the NFkb knockdown at the gene and protein level, suggests that the genetically unmodified SCs are not contributing substantial additional myonuclei.”

6. It has been previously documented that NF-κB is not the only catabolic signal that activates ubiquitin-proteasome and lysosomal autophagy. Foxo3 can regulate both UPS and autophagy. Moreover, there are several E3 ligases other than MAFbx1 and Murf1 that play a role in muscle atrophy conditions.

We agree completely on this point. The fact that we see increased protein ubiquitination at week 1 post tenotomy that is unaffected by IKKb knockdown, supports this contention that there are other regulators. We have now added a more detailed discussion of these potential mediators.

Discussion:

“Notably, Foxo3 regulates autophagy/lysosomal and ubiquitin/proteasomal degradation (Zhao et al., 2007) and this master regulator was not assessed in detail here. Additionally, a number of E3 ubiquitin ligases other than *Murf1* and MAFbx regulate muscle atrophy (Ye *et al.*, 2007; Nagpal *et al.*, 2012; Hindi *et al.*, 2014). The complex interplay between these pathways warrants further investigation.”